# Common irrigation drivers of freshwater salinisation in river basins worldwide

Josefin Thorslund [1,2 ✉], Marc F. P. Bierkens [2,3], Gualbert H. P. Oude Essink [2,3], Edwin H. Sutanudjaja[2] & Michelle T. H. van Vliet [2]

Freshwater salinisation is a growing problem, yet cross-regional assessments of freshwater salinity status and the impact of agricultural and other sectoral uses are lacking. Here, we assess inland freshwater salinity patterns and evaluate its interactions with irrigation water use, across seven regional river basins (401 river sub-basins) around the world, using long-term (1980–2010) salinity observations. While a limited number of sub-basins show persistent salinity problems, many sub-basins temporarily exceeded safe irrigation water-use thresholds and 57% experience increasing salinisation trends. We further investigate the role of agricultural activities as drivers of salinisation and find common contributions of irrigation-specific activities (irrigation water withdrawals, return flows and irrigated area) in sub-basins of high salinity levels and increasing salinisation trends, compared to regions without salinity issues. Our results stress the need for considering these irrigation-specific drivers when developing management strategies and as a key human component in water quality modelling and assessment.

[1] Department of Physical Geography and the Bolin Centre for Climate Research, Stockholm University, Stockholm, Sweden. [2] Department of Physical Geography, Utrecht University, Utrecht, The Netherlands. [3] Unit Subsurface and Groundwater Systems, Deltares, The Netherlands. ✉email: josefin.thorslund@natgeo.su.se

Freshwater salinisation is a major water quality problem, affecting both surface- and groundwater systems, with impacts on agricultural production, sectoral water use, biogeochemical cycling, water quality regulation, as well as human and ecosystem health[1–3]. Although increasing salinity is a growing threat in many regions[4], salinity issues have received limited attention compared to other global water quality issues, such as eutrophication, pesticides and metal pollution[5]. Particularly, more large-scale (cross-regional) research on inland surface water salinity levels and change has been called for, as well as long-term salinity effects on human water use sectors, including the agricultural sector[6,7]. Evaluating salinity impacts for irrigation water use is particularly relevant as it is the largest water use sector globally and it has the most stringent salinity threshold compared to other sectors[8]. This sector is vulnerable to salinity issues within the whole water-soil-crop system and salinisation is seen as a major threat to the sustainability of irrigated agriculture and food production[9].

In addition to quantifying the status and impact of freshwater salinisation, a better understanding of its large-scale drivers (Fig. 1) is needed[10]. Evaluation of large-scale drivers in coastal regions has been well-studied[11] and includes saltwater intrusion caused by excessive groundwater pumping[12], extreme hydrological events[13], tidal flows and/or storm surges[14]. These processes are likely to increase under climate change and relative sea-level rise[15]. However, the need to also assess and manage drivers of inland salinisation, both in arid and humid regions, is becoming more and more pressing[16]. In addition to hydroclimatic drivers[17], relevant human drivers to inland freshwater salinisation include road salt application[18], mining[19] and agriculture[20]. Among these human drivers, the impact of road salts on rising salinity levels has been relatively well-quantified and reviewed across multiple inland surface water bodies and regions[18,21]. With regards to agricultural drivers, there are several studies at local and regional scales[22,23], including

raising saline groundwater due to irrigation withdrawals[24], or spreading of salt-containing fertilisers to the groundwater and/or surface water system[25]. However, there is a lack of systematic assessments of the impact of agricultural activities on surface water salinisation at cross-regional to global scales.

Here, we assess freshwater salinisation and its relation to agricultural activities, both in terms of (i) impacts of salinity levels for irrigation water use, as well as (ii) contributions of agricultural drivers to freshwater salinisation across regional scales. Specifically, we quantify salinity levels, trends and drivers in 401 river sub-basins (hereafter referred to as sub-basins), across seven regional river basins around the world (Fig. 2). These regions, which span different hydroclimatic and geographical regions and varying anthropogenic impacts, are: Mississippi (North America), Ebro, Danube, Rhine (Europe), Orange (Africa), Mekong (Asia) and Murray-Darling (Australia). Within these regions, we delineate the sub-basins from all river monitoring locations with a minimum of 15 years of monthly salinity data within the period 1980–2010 (Supplementary Figs. 1 and 2). We compute monthly mean salinity levels of stations from over 400,000 salinity observations (electrical conductivity; EC), synthesised from a new open-access global database[26]. We then classify the sub-basins into different salinity levels, based on international threshold values for irrigation water use[27,28] and assess salinity trends over the 1980 and 2010 period. We further acquire global-scale open data of agriculturally relevant variables for evaluating their contributions to observed salinity levels, as well as some additional hydroclimatic and geographic variables that are expected to influence salinity levels. Sub-basin averages of 26 driver variables (Table 1 and Supplementary Section S2) are calculated and their relationship across salinity levels and trends are assessed, using both statistical and machine-learning (random forest (RF)) methods[29]. Our results show that irrigation-specific activities (particularly irrigation water withdrawals, return

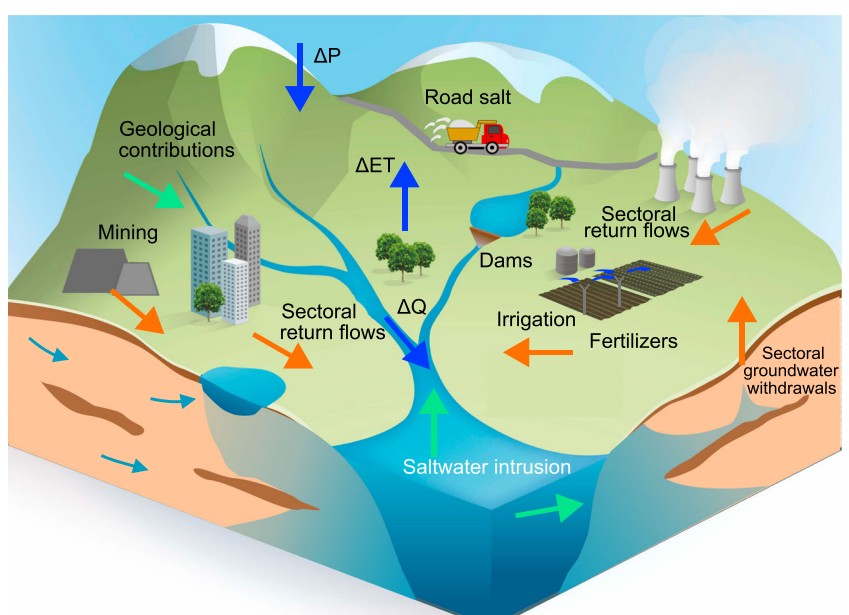

**Fig. 1 Potential drivers of freshwater salinisation within a hydro(geo)logical landscape.** The drivers are categorised according to hydroclimatic (blue arrows), geographic (green arrows) and human (orange arrows) drivers. Examples of human drivers include polluted return flows from agricultural, domestic, and industrial activities (including mining), as well as road salt applications and saltwater intrusion caused by excessive groundwater withdrawals. Geographic drivers for instance include geological weathering products, atmospheric salt deposition and natural saltwater intrusion through seawater inundation caused by fresh-saline water density difference effects, tides, storm-surges and even long-term transgressions. Hydroclimatic drivers may impact salinisation through changes in discharge (ΔQ) and resulting dilution capacity, as well as through increasing evapo-concentration under increasing evapotranspiration (ΔET) and/or decreasing precipitation (ΔP) or discharge (ΔQ).

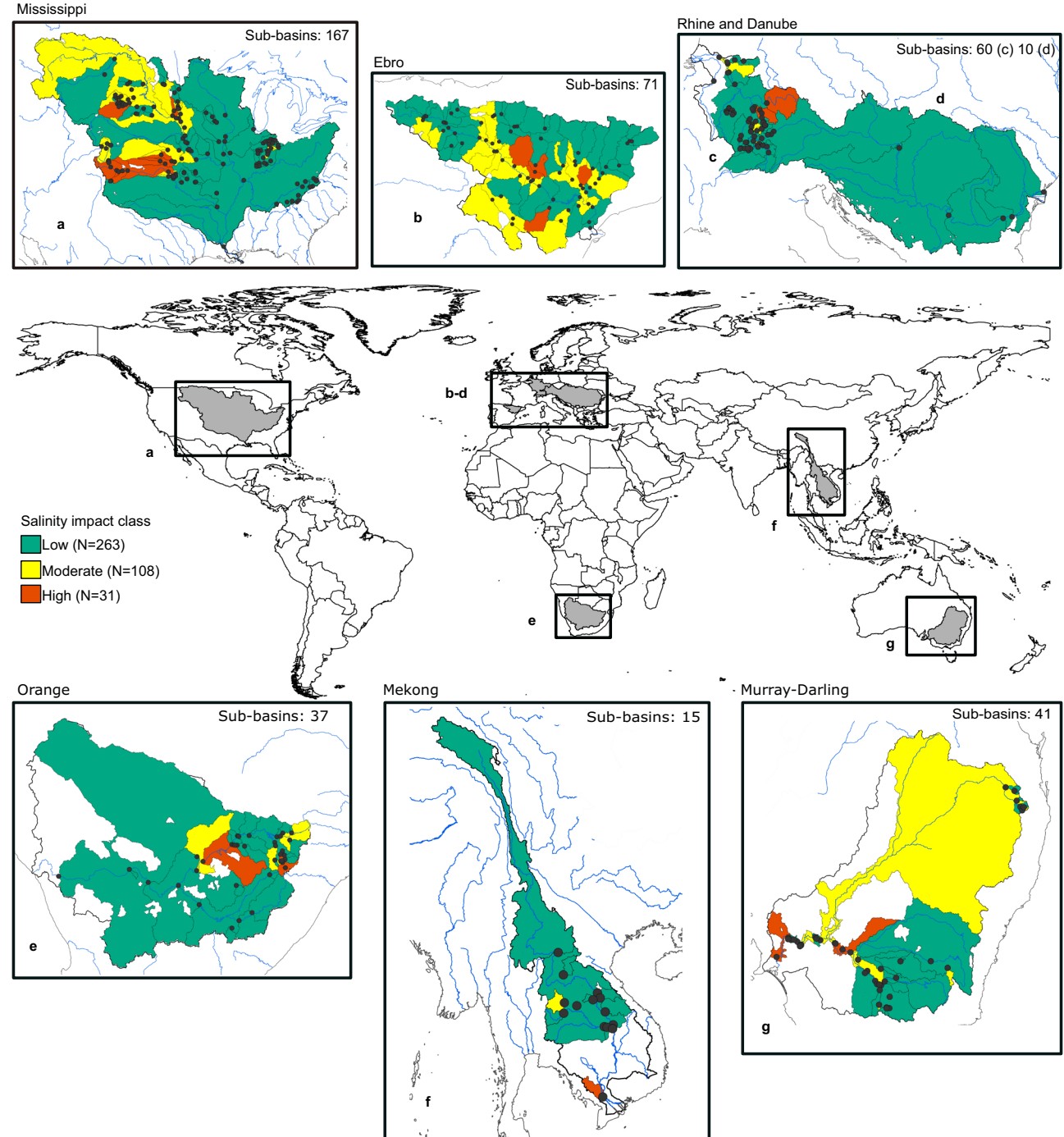

**Fig. 2 Long-term average salinity levels across the studied river sub-basins.** Overview of the seven selected regional river basins and associated delineated 401 sub-basins from monitoring locations (black dots), with number of sub-basins of each regional river basin given in each zoomed panel. The salinity (EC) data availability criteria per monitoring station is at least 15 years of monthly data over the 1980–2010 period. The full distribution of salinity measurements per station is given in Supplementary Fig. S1.2. Each sub-basin is classified into different salinity impact classes, based on thresholds for irrigation water use, with Low (EC < 700 μS cm$^{-1}$; green colour), Moderate (<700–1500 μS cm$^{-1}$; yellow colour) and High (>1500 μS cm$^{-1}$; orange colour) classes and associate total number of sub-basins within each class (N), based on long-term annual average EC over the 1980–2010 period. Sub-basins in white did not meet the data criteria of stations with a minimum of 15 years of monthly salinity data within the study period.

flows and irrigated area) are significantly elevated in sub-basins of high salinity levels and in regions that have experienced increasing salinisation, compared to regions without salinity issues. The identified significance of irrigation-specific drivers offers guidance for the development of salinity-specific monitoring programs and for considering irrigation as a key human component in water quality assessments and modelling.

## Results

**River salinity levels and threshold exceedance.** Overall, a majority of the investigated sub-basins show low long-term average salinity levels, with 65 % of the total 401 sub-basins occurring within the Low salinity impact class, which is below the threshold value for irrigation water use (<700 μS cm$^{-1}$) (Fig. 2). In contrast, 35% of the sub-basins have long-term

**Table 1 Information on all main driver variable groups and their respectively sub-categories included in this study.**

| Driver variable (abbreviation; unit)[a] | Dataset | Dataset resolution | Source |
|---|---|---|---|
| Temperature (T, °C) | CRU TS4.03 | Globally gridded (0.5°), at monthly temporal resolution (1980–2010) | https://catalogue.ceda.ac.uk/uuid/10d3e3640f004c578403419aac167d82 |
| Precipitation (P; m month$^{-1}$) | PCR-GLOBWB 2[73] | Globally gridded (5×5 arc-minute), at monthly temporal resolution (1980–2010) | https://www.geosci-model-dev.net/11/2429/2018/ |
| Potential evapotranspiration (PET; m month$^{-1}$) | PCR-GLOBWB 2 | Same as above | Same as above |
| Actual evapotranspiration (AET; m month$^{-1}$) | PCR-GLOBWB 2 | Same as above | Same as above |
| Evaporative ratio (−): (i) PET/P and (ii) AET/P | Estimated from P, PET, AET | Same as above | Same as above |
| Discharge (Q; m$^3$ sec$^{-1}$) | PCR-GLOBWB 2 | Same as above | Same as above |
| Sectoral water withdrawal (m month$^{-1}$): (i) irrigation (Irr. ww), (ii) non-irrigation sectors (domestic, industrial and livestock combined), (Non-irr. ww) | PCR-GLOBWB 2 | Same as above | https://www.geosci-model-dev.net/11/2429/2018/ |
| Return flows (m month$^{-1}$) for: (i) irrigation (Irr. rf) (ii) non-irrigation sectors (Non-irr. rf; sum of domestic, industrial and livestock sectors) | PCR-GLOBWB 2 | Same as above | Same as above |
| Dams: (i) total dam capacity (Dam storage; Mm$^3$), (ii) number (Nr. Dams; −) (iii) ratio of dam capacity to sub-basin area (Norm. dam storage; −), (iv) ratio of dam area to sub-basin area (Norm. dam area; −) | Global Reservoir and Dam (GRanD), v1[74,75] | Global coverage | https://sedac.ciesin.columbia.edu/ |
| Land-use (m$^2$): (i) Total cropland area (Tot. cropland) and (ii) Ratio irrigated area to sub-basin area (Norm. irr. area) | (i) NOAA-HYDE[76,77], (ii) MIRCA2000[78] | (i) Globally gridded (0.5°) at annual temporal resolution (1980–2010), (ii) Globally gridded (0.5°) at year 2000 | (i) https://data.nodc.noaa.gov/cgi-bin/iso?id=gov.noaa.ncdc:C00814# (ii) https://www.uni-frankfurt.de/45218023/MIRCA |
| Fertiliser application (tons) of: (i) Nitrogen (N. total) (ii) Phosphorous (P. total) and (iii) Total (sum of (i) and (ii); Tot. Fert.) | Nitrogen and phosphorus use for global agriculture production during 1900-2013[79] | Global gridded (0.5°), monthly resolution 1980–2010 | 10.1594/PANGAEA.863323 |
| Soil salinity (dS m$^{-1}$) of (i) top layer (0–20 cm; EC top soil), (ii) sub soil layer (20–40 cm; EC sub soil) and (iii) its average (EC soil aver.) | WISE30sec[80] | Globally gridded, at 30 arc-seconds | https://data.isric.org/ |
| Elevation (m.a.s.l.) | Void-filled, hydrologically corrected DEM | Globally gridded, at 15 arc-sec resolution | https://www.hydrosheds.org/ |
| (i) Actual distance from coast (Dist. coast; km), (ii) Relative distance from coast (Rel. dist. coast; −) | (i) Coastline vector map from Natural Earth, (ii) calculated from (i) and elevation | Global coverage | https://www.naturalearthdata.com |

[a]Detailed descriptions of processing steps for each driver variable are included in the supplementary information (Supplementary Section S2).

average salinity levels exceeding this threshold value, with the majority ($n = 108$ out of 401) within the Moderate salinity impact class (700–1500 μS cm$^{-1}$) and a smaller subset ($n = 31$) within the High salinity impact class (>1500 μS cm$^{-1}$). Considering the spatial context, our results show both within and cross-regional heterogeneity of salinity levels, with the occurrence of sub-basins of all salinity impact classes in each of the regional river basins. The overall largest variability across sub-basins is seen in the Mississippi, Ebro and Murray-Darling river basins, while the Mekong and the Danube river basins show the overall lowest variability across sub-basins (while dominated by the Low salinity impact class).

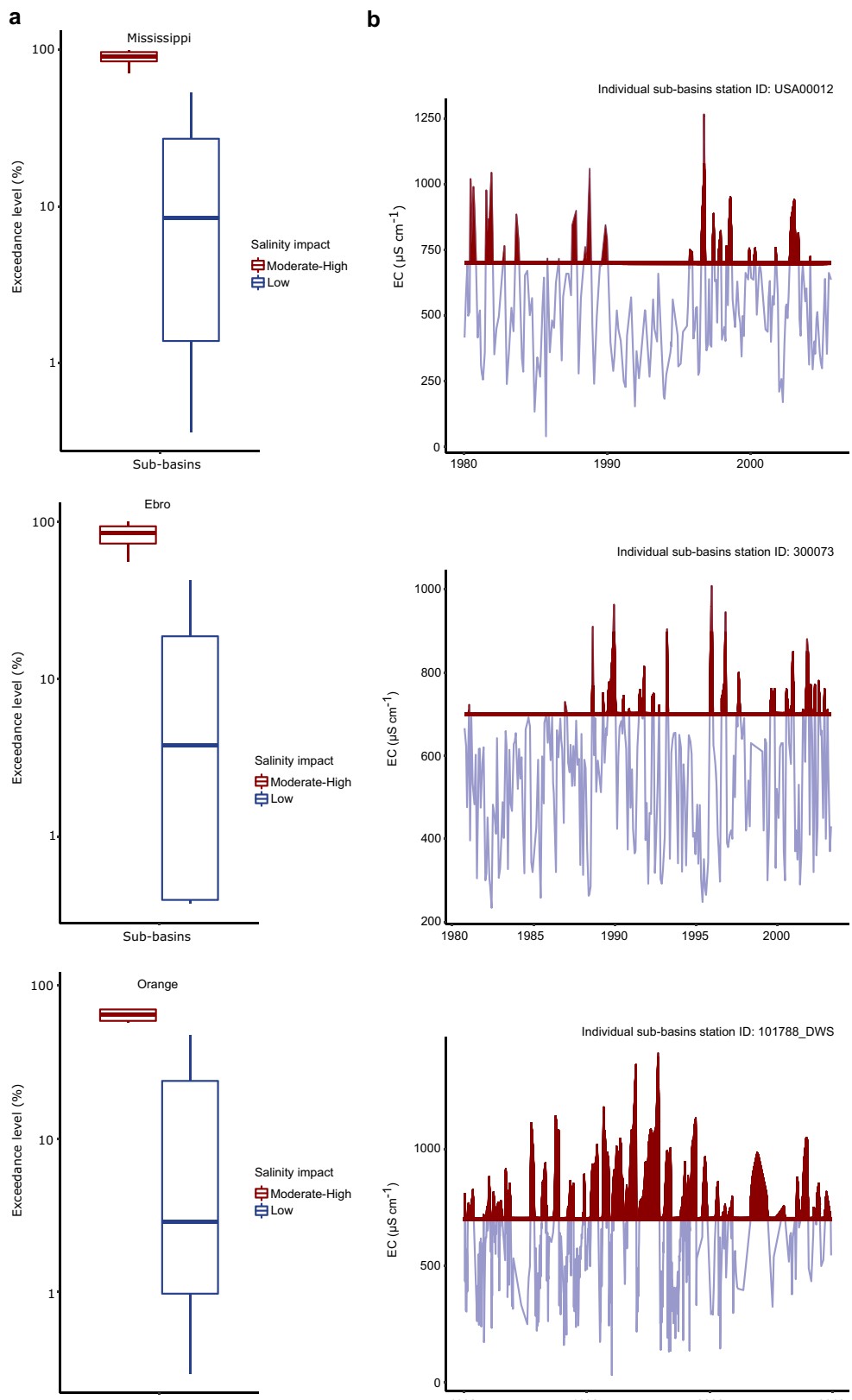

**Fig. 3 Distribution and timeseries of sub-basin salinity threshold exceedance.** Panel **a** shows three examples of the distribution of sub-basin salinity exceedance levels (% of months exceeding the threshold 700 μS cm$^{-1}$ compared to total number of monthly measurements for each sub-basin) within the Mississippi, Ebro and Orange regional river basins, grouped over sub-basins within the Low salinity impact class (i.e., sub-basins with long term annual average salinity values below 700 μS cm$^{-1}$; blue boxplots), and sub-basins within the Moderate-High salinity impact class (i.e., long-term salinity levels above 700 μS cm$^{-1}$; red boxplots). Panel **b** shows monthly timeseries of one individual sub-basin station from the Low salinity impact class (selected from the longest timeseries available), from each river basin example in **a**, with salinity levels exceeding the threshold highlighted in red.

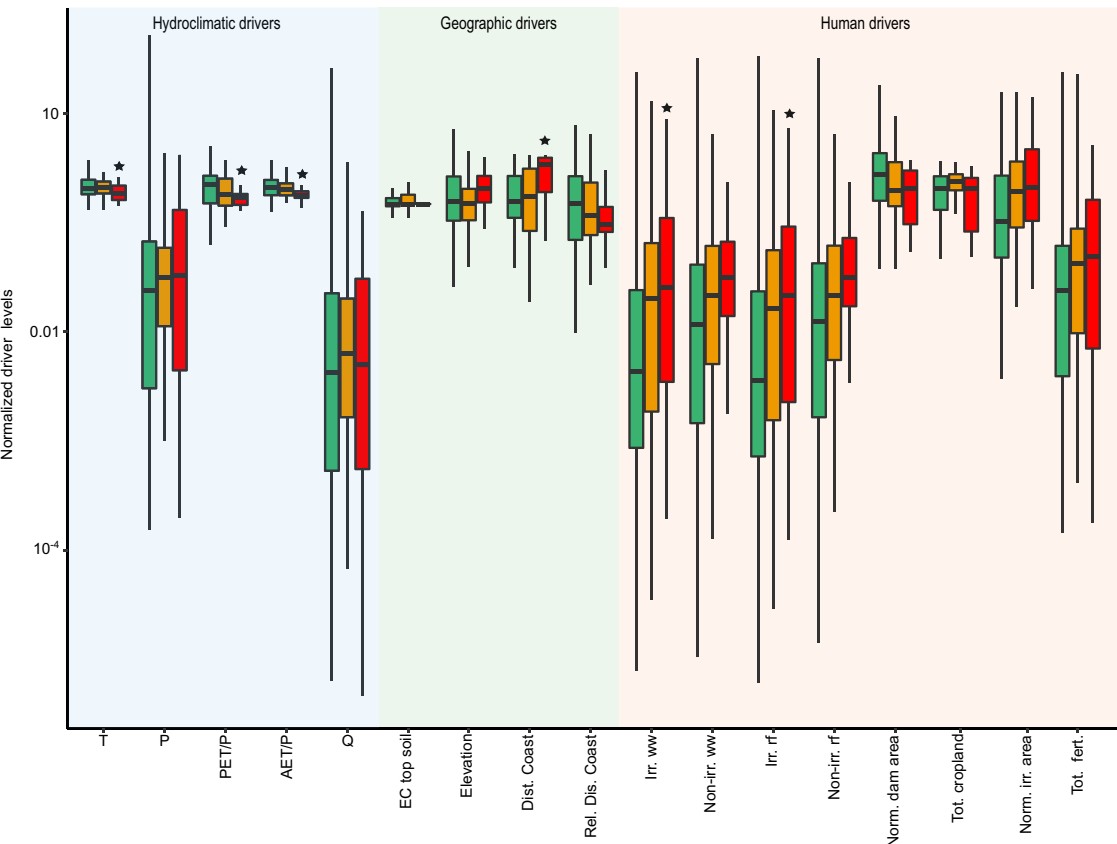

**Fig. 4 Contributions of hydroclimatic, geographic and human drivers across sub-basin salinity impact classes.** Distribution of driver levels across sub-basin salinity impact classes, for a selection of 17 out of the total 26 considered driver variables. The selection includes all driver categories (as listed in Table 1), but where multiple variables within the same category exists (e.g., for soil salinity), only one variable was included. The salinity impact classes are based on groups of sub-basins with Low (EC < 700 µS cm$^{-1}$; green boxplots), Moderate (EC = 700–1500 µS cm$^{-1}$; yellow boxplots) and High (EC > 1500 µS cm$^{-1}$; orange boxplots) salinity levels, classified from long-term annual average values (as illustrated in Fig. 2). The selected drivers are plotted along groups of (i) hydroclimatic, (ii) geographic and (iii) human (agricultural-related) drivers on the x-axis, and their normalised levels on the y-axis. The values were normalised by dividing each sub-basin specific driver value by the group-average value, for each driver and then grouped over each salinity impact class. Full driver names and original units are given in Table 1. Boxplot statistics include the median (vertical thick black lines), interquartile range (IQR: 25th percentile; Q1 and 75th percentile; Q3) and whiskers (confidence interval of Q1 − 1.5∗IQR to Q3 + 1.5∗IQR). The stars (*) over the boxplots indicate where there are statistically significant differences in the driver contributions between the Low-High salinity impact classes (all summary statistics are included in Supplementary Table 1).

Compared to these long-term average levels, we also analyse sub-basin exceedance frequency of salinity values above the 700 µS cm$^{-1}$ irrigation threshold, both within and across regions (i.e., the fraction of months compared to total that each sub-basin exceeds the salinity threshold). We separate the effects to show the exceedance level of all sub-basins within the Low salinity impact class (i.e., sub-basins with long term annual average salinity values below 700 µS cm$^{-1}$), compared to all sub-basins within the Moderate-High salinity impact class (i.e., long-term salinity levels above this threshold). Examples from the Mississippi, Ebro and Orange basins highlight that even when long-term salinity levels are below the 700 µS cm$^{-1}$ threshold (i.e., Low salinity impact class), sub-basin exceedance still occur on shorter timescales (blue boxplots of Fig. 3a) and there can be large intra-annual variability in salinity levels (Fig. 3b). We see similar results considering all regions, with an average sub-basin exceedance level of 33% (Supplementary Fig. 3a), but with highly variable results (sub-basin exceedance levels of between 0 and 58%; Supplementary Fig. 3b). Such intra-annual threshold exceedance can have important implications for water use management, as these water resources may not always be of suitable salinity level for irrigation water use purposes.

**Driver contributions across salinity levels**. To investigate the contributions of the selected set of driver variables to observed salinity levels, we quantify long-term (i) (agriculture-related) human (ii) hydroclimatic and (iii) geographic variables over each sub-basin, and compare their contributions across the different salinity impact classes (Low, Moderate, High). To statistically assess the different contribution of each driver across observed salinity impact classes, we perform the Wilcoxon rank sum test. This test identifies if there is a statistically significant shift in the distribution of each driver variable between sub-basins of different salinity impact classes (Low-High, Low-Moderate, Moderate-High), at the 95% significance level, i.e., $p < 0.05$ (see Methods for further details).

Our results show higher contributions of (human) agricultural drivers in sub-basins of Moderate to High salinity levels (Fig. 4; yellow and orange boxplots), compared to those of Low salinity levels (Fig. 4; green boxplots). These results are particularly strong for irrigation-related variables, with significantly higher levels of irrigation water withdrawals (Irr. ww) and irrigation return flows (Irr. rf) in sub-basins within the High, compared to the Low salinity impact class (Fig. 4; green-orange boxplots, and Wilcoxon test results in Supplementary Table 1; Low-High pair comparison). Our results also show the contribution of agricultural land-

use, with higher fractions of irrigated area (Norm. irr. area), total cropland area (Tot. cropland) and fertiliser use (Tot. fert.) in sub-basins of the Moderate, compared to the Low salinity impact class (Fig. 4; green-yellow boxplots, and Supplementary Table 1, Low-Moderate class comparison), as well as to a lower extent non-irrigation human water withdrawal and return flows (Non-irr. ww; Non-irr. rf; Fig. 4 across boxplot levels, and Supplementary Table 1). We also find evidence of similar agricultural land-use contributions with increasing salinity levels within most regional river basins (Supplementary Fig. 4). Specifically, sub-basins of higher salinity levels within the Mississippi, Ebro, Orange and Mekong basins have higher contributions of irrigation water withdrawals and return flows (Irr. ww; Irr. rf; Supplementary Fig. 4a, b, e, f). There are also within-regional effects of higher cropland (Tot. cropland; Supplementary Fig. 4b, c, e, f, g) and relative irrigated area (Norm. irr. area; Supplementary Fig. 4a, b, e, f, g) with increasing salinity levels. In contrast, higher salinity sub-basins within the Rhine and Murray-Darling are dominated by a higher contribution of non-irrigation (i.e., domestic, industrial and livestock) water withdrawals and return flows (Non-irr. ww; Non-irr. rf; Supplementary Fig. 4c, d).

Surprisingly, for the hydroclimatic drivers, although we observe significant differences in evaporative ratios (PET/P, AET/P) and temperature (T) between salinity impact classes, the contribution of all these drivers decrease with increasing salinity levels (Fig. 4; direction across from green to orange boxplots). Such changes are opposite to expectations based on evapo-concentration effects (i.e., increasing salinity with increasing evaporation). It is also generally expected that the impact of irrigation in high salinity regions would correspond to more arid regions, which requires more irrigation. To further investigate this unexpected correlation between salinity and aridity, we explore relations between aridity and irrigation within each sub-basin, including irrigated area, irrigation consumption and irrigation water withdrawals. We find that all these irrigation-specific parameters have a higher contribution in sub-basins of lower aridity (lower aridity index; PET/P), compared to higher aridity regions (Supplementary Fig. 5). These results support our finding of the contribution of irrigation drivers in high salinity sub-basins, and explain the inverse relation between salinity levels and aridity, since irrigation area and water use are elevated in less arid regions. A likely explanation to why we observe lower irrigation water use with increasing aridity could be due to water limitations in the more arid regions (172 of our investigated sub-basins occur in dry sub-humid to arid regions). While crop irrigation water demands are expected to be higher in more arid catchments, it is also more likely that the demand cannot be met due to a lack of available water resources (i.e., actual irrigation withdrawals and consumption is thus lower). Another possible explanation to the inverse relationship between irrigation and aridity could be due to irrigation-climate interactions. Recent studies have shown that irrigation can lead to increasing soil moisture and relative humidity, which in turn, reduces atmospheric aridity[30]. Due to the land–atmospheric coupling processes described above, irrigated regions across the world can therefore cause local/regional atmospheric cooling, which may reduce aridity[31]. Such effects could also contribute to observed relationships between lower aridity and increasing irrigation, which are in line with our findings of higher salinity in less arid regions.

To further investigate the robustness of the driver analyses results, we also test for significance between driver contributions across two new salinity impact classes, based on salinity levels above ($N = 139$) and below ($N = 262$) the salinity threshold level of 700 µS cm$^{-1}$ for irrigation water use. This alternative grouping still allows for comparison between sub-basins with and without salinity issues for irrigation water use, but increases the sample

size of the sub-basins with elevated salinity levels (above; >700 µS cm$^{-1}$). Our results for this sub-group analysis also agree with previous quantifications, showing again the high contribution of irrigation-specific and agricultural land-use drivers (Irr. ww, Irr. rf, Tot. cropland, Norm. irr. area, Tot. fert.), as well as lower evaporative indexes in salinity-affected regions (Supplementary Table 2). Although our focus is on evaluating the impact of agricultural-related human drivers, we perform additional analyses to quantify possible impacts of mining and road salt use on salinity levels, to test the robustness of our results. We use a new global gridded dataset of mining areas[32], to evaluate the contribution of mining area (total and normalised) within all sub-basins, across salinity impact levels. For road salt, we estimate annual average application (in pounds) within available years of our study period (1992–2010)[33] for all sub-basins within the Mississippi river basin as a case study ($N = 167$). We focus on this region, due to due to limitations of global datasets and the lack of relevance of road salt in many of our considered regions (located in climate zones with snow-free winters, where road salt application is not present/limited). We did not find any significant contributions of road salt or mining in sub-basins within the High or Moderate salinity impact class (Supplementary Fig. 6).

**River salinity trends**. Salinity trends of each sub-basin were then quantified, based on Mann-Kendall trend tests[34,35] and Sen's slope analyses[36], considering monthly timeseries over the 1980–2010 period. We found that 57% (229 out of 401) of all investigated sub-basins exhibit increasing basin-average salinity trends over the study period (with 32% of total sub-basins showing statistically significant increasing trends; $p < 0.05$) (Fig. 5a, b). In contrast, 43% (172 of 401) of the sub-basins show decreasing basin-average salinity trends over the considered 30-year period (with 24% of total sub-basins being statistically significant). Most regions show spatial heterogeneity across sub-basins, with the presence of both salinisation and decreasing salinity trends, sometimes in close proximity (see e.g., clusters of opposite trends within Mississippi, Ebro and Murray-Darling), whereas some regional river basins (e.g., Orange) show more consistent trends throughout the sub-basins (Fig. 5a). Overall, the Murray-Darling (78%) and Danube (73%) river basins are dominated by decreasing salinity trends, whereas the Mississippi and Orange basins have the relatively largest number of sub-basins experiencing salinisation (73% respectively 63% of their total sub-basins show increasing trends; Fig. 5). A large proportion of the positive salinity trends in the Mississippi and Orange basins are statistically significant (Fig. 5b), highlighting a stronger within-regional degree of salinisation compared to other regions.

**Driver contributions across salinity trends**. We used the Random Forest (RF) approach, together with the conditional permutation importance (CPI)[37] method, to examine the relative importance of the considered drivers to predicting salinity levels in sub-basins of contrasting salinity trends (increasing vs. decreasing trends; see Methods and Supplementary Section S4). The RF model performs slightly better ($R^2 = 0.62$; Supplementary Table 4) in predicting salinity levels for sub-basins experiencing salinisation (i.e., the group of sub-basins with increasing long-term salinity levels), compared to sub-basins experiencing decreasing salinity trends ($R^2 = 0.51$; see Supplementary Table 4 for all accuracy parameters). In terms of driver importance evaluation, we get the same top 15 predictors regardless of chosen CPI threshold, suggesting robustness of this method within the chosen threshold range (see Methods for specifics). For sub-basins experiencing salinisation, our

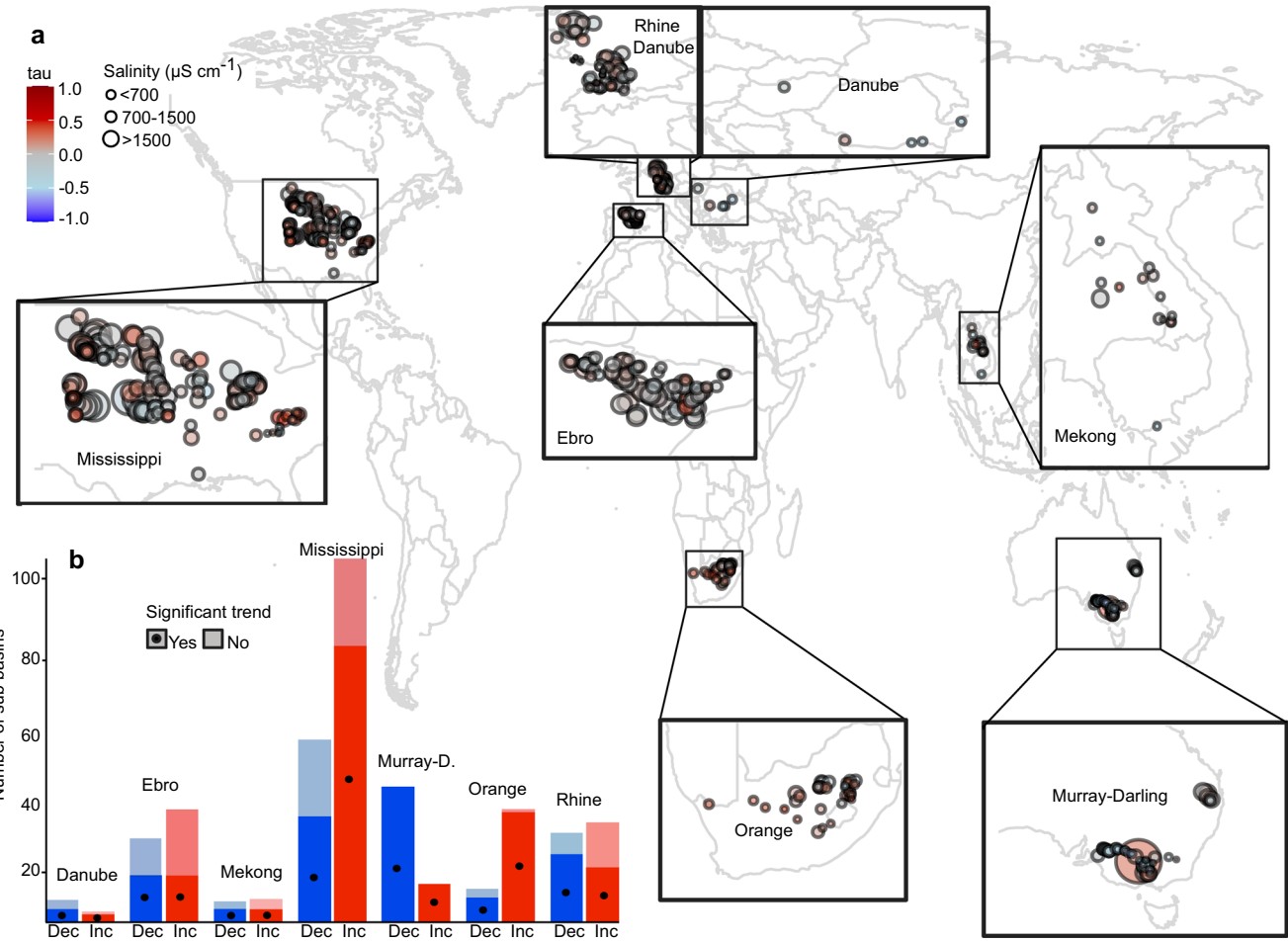

**Fig. 5 Salinity levels and trends across and within the seven regional river basins.** Panel **a** shows an overview of long-term salinity levels (size of circle) and Mann-Kendall trend test results (colour of circle) for each sub-basin monitoring location over the study period (1980–2010). Salinity trends are expressed in tau-values, ranging from −1 to 1, where negative values (in blue) indicate decreasing salinity trends and positive values (in red) indicate increasing salinity trends. The stacked bar-plot in panel **b**, shows the distribution of the number of sub-basins with increasing (red) vs. decreasing (blue) salinity trends within each regional river basin and whether the trend is statistically significant (darker colour with dot) or not (lighter colour), estimated at the 95% confidence level (*p* value < 0.05).

results clearly show that irrigated area (Norm. irr. area; Fig. 6a), plays the most significant role in achieving prediction accuracy of the RF model, out of the 26 included driver variables. This is identified by its higher CPI accuracy importance score compared to the other variables (unitless; a higher score means a higher relative contribution compared to other variables). Other top predictors include landscape position (Dist. coast; Rel. dist. coast; Elevation) and soil salinity (EC top soil), as well as, from the human driver variables; irrigation water withdrawals and return flows (Irr. ww, Irr. rf).

For sub-basins with decreasing salinity trends, distance from the coast plays the most significant role in achieving prediction accuracy, suggesting this geographic variable is important to consider in all salinity predictions. In contrast, for human driver variables, non-irrigation return flows and withdrawals emerges as top predictor variables (Non-irr. rf, Non-irr. ww; Fig. 6b) for sub-basins with decreasing salinity trends. This suggests that other sectoral activities, rather than irrigation activities, may be more important to consider when predicting salinity levels in areas that are not experiencing salinisation. The gap is however less between the top driver and the other variables here (Fig. 6b) compared with increasing salinisation locations (6a), suggesting a lower distinctive individual contribution from any driver among sub-basins experiencing decreasing salinity trends.

The relative importance of irrigation-specific drivers in sub-basins that have undergone increasing salinisation, but not in sub-basins with decreasing salinity trends, are in line with our earlier analyses (Fig. 4), and also highlights the contribution of irrigation activities in regions with increasing salinity trends. In line with earlier analysis, we also included mining and road salt as predictor variables in the RF and CPI-analysis, to predict salinity levels in sub-basins experiencing salinisation. Mining showed a very low CPI accuracy importance score compared to the other variables, which indicates its lack of relevance for predicting salinity levels within the here studied regions (Supplementary Fig. 8a, b). For the Mississippi case study example, road salt was more important than mining, but was still only the 14th highest ranked variable in this region, scoring lower than all irrigation-related parameter (Supplementary Fig. 8a, b). To further understand the absolute contributions of the different drivers between sub-basins of increasing and decreasing salinity levels, more detailed analysis of actual driver levels and/or trends should be considered. For example, irrigation return flows (Irr. rf) emerge among the top 15 predictors for both increasing and decreasing salinity trends (Fig. 6a, b). This means that this variable is important to consider when predicting salinity levels, regardless of the underlying salinity trend. However, actual driver impact level must be related to further analyses, for instance combined

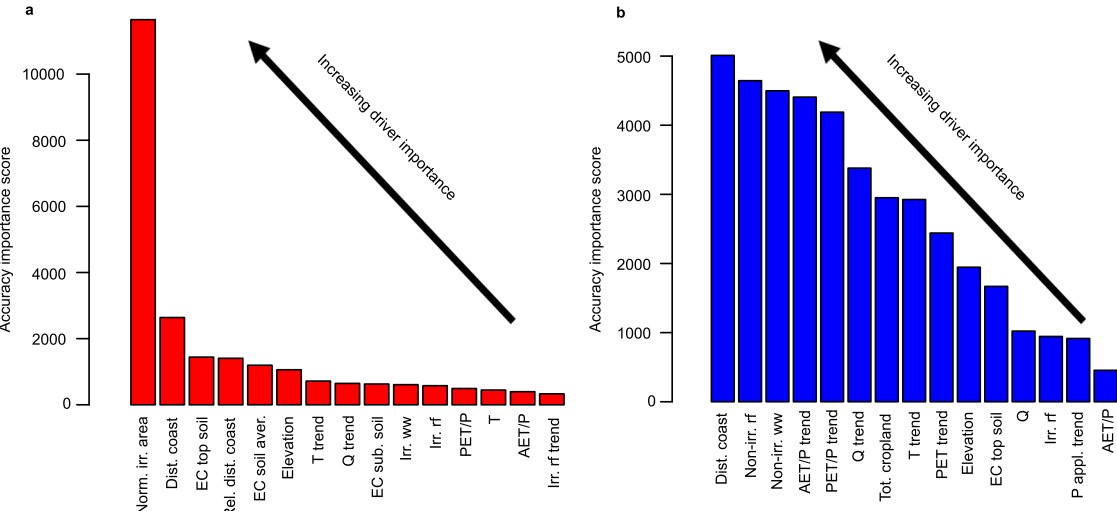

**Fig. 6 Relative driver contributions for predictions of salinity levels across trends.** Bar charts show the relative importance of driver variables in predicting salinity levels in sub-basins with (**a**) increasing salinity trends (red bars) and (**b**) decreasing salinity trends (blue bars), using the conditional permutation importance method within the Random Forest approach. The accuracy importance score (y-axis; unitless) represent the relative importance of each predictor variable in achieving the prediction capacity of the Random Forest model (note the different y-axes in panel a and b and that the absolute values of the scores should not be interpreted, only the relative scores between drivers).

with correlations assessment (Fig. 4) or other statistical analyses. The results of the RF and the CPI analyses are valuable tools for moving away from "black-box" type predictions, by identifying variables that are critical for successful salinity predictions. Using these methods in regions experiencing contrasting salinity trends is an important first step for improving our understanding of relations between salinity levels, trends and drivers.

## Discussion

This study provides new data-driven insights on cross-regional salinity levels, trends and drivers in a large selection of river sub-basins worldwide. Many local to regional studies have suggested that freshwater salinisation is occurring at an increasing rate and at expanding scale[4–6,10]. Using long-term publicly available salinity monitoring data[26], we here expand on the spatial scale of previous findings, by showing that a majority of quantified sub-basins indeed experience increasing salinity trends. However, our results also reveal large spatial heterogeneity in the direction of trends, both within and across (regional) river (sub-) basins. We also found that many sub-basins which did not exhibit persistent salinity problems, still temporarily exceeded salinity irrigation water-use threshold levels. This suggests that problems of freshwater salinisation might fluctuate highly, both within and across seasons[38], which could have important implication for intra-annual irrigation water use. In addition, salinity water use constraints are underestimated if quantified at the annual to long-term scales, highlighting the need for further studies to account for seasonal aspects with regards to both salinity levels and trends.

Irrigating with saline water could severely affect crop yields, due to long-term soil salinisation build-up, waterlogging and reduced water availability for crops when salinity increases[39]. Yield reductions of between 10 and 25% have been shown to affect saline sensitive crops (e.g., beans, carrots and lettuce) when irrigating with saline water of around 1000 µS cm$^{-1}$ (i.e., salinity levels within the Moderate and High salinity impact classes of this study)[40]. Although salt-tolerant crops (such as halophytes) exist and may provide a suitable adaptation strategy to high salinisation for some regions, the large-scale potential of saline irrigation is still an underexplored research area[41]. In addition, the risks of

higher yield reductions increase over time, due to soil salinity build-up[42]. Given the global relevance of the irrigation water use sector, our assessment highlights both vulnerable areas in need of further salinity assessment and management, as well as regions where salinity levels are not (yet) an issue. Additional research, focusing on agricultural production impacts, considering for instance crop-specific salinity thresholds over the seasons and yield analyses, would add valuable details to our large-scale analysis. More generally, although our focus has been on evaluating salinity levels with regards to limitations for irrigation of most crops, our results and salinity dataset could be used for other impact assessments. This includes other human water uses, such as increasing water scarcity for the domestic and industrial sectors[43], as well as human health[44]. Also, in terms of ecological assessments, high freshwater salinity levels can have multiple effects on biota, with impacts on species richness, abundance and functional traits, as well as on ecosystem processes[6,45]. However, it is worth noting that naturally occurring saline aquatic systems, especially in arid regions, also provide important ecosystem services and valuable habitats for salt adapted species[46]. Therefore, high salinity levels do not always result in ecosystem degradation, but salinity levels must be evaluated with regards to underlying conditions and specific impact assessments. Further cross-regional assessments with regards to these various aspects could add valuable knowledge to the more wide-ranging impacts of the here quantified salinity levels.

In addition to being affected by high salinity levels, our analyses show cross-regional effects of the contribution of agricultural, and particularly irrigation-specific activities (irrigation water withdrawals, return flows and irrigated area), in sub-basins of elevated salinity levels and increasing salinity trends, compared to regions with lower or no salinity issues. These results both agree with, and increase the spatial extent of previously reported effects of agricultural activities on salinisation[21,22]. For example, a study using a similar RF approach as here and considering drivers to river salinisation across the US showed that the largest salinity increases were impacted by a combination of agriculture (% crop land cover), mining and groundwater pumping[47]. Irrigation-specific practices have also been shown to increase salinity of surface waters, as salts evapoconcentrate from the application of

irrigation water to agricultural fields[48]. This is a particularly (well-known) reported problem in arid and semi-arid regions where large water volumes are needed for crop production. Since all salts in the irrigation water are not absorbed by the crops, salt may concentrate in the soil, which can lead both to soil degradation and spreading of salts to surface and groundwater resources[49]. Our results however show that these problems are not limited to arid regions, but that irrigation activities could also significantly contribute to salinity issues in more humid regions. These results highlight the need for more studies on the interactions between salinity and irrigation practices, water availability and soil characteristics beyond arid regions.

Our results are based on both classical statistical and machine-learning (random forest (RF)) methods, which have been used successfully in a range of environmental studies, including salinity issues[50]. The inclusion of the CPI method[37] allows more reliable interpretation of driver contributions, which has earlier been a main issue within RF modelling analyses[51]. Additionally, machine-learning methods offers potential for predicting salinity patterns in areas where empirical measurements are few or lacking, as well as for usages with other salinity proxies, including remote sensing datasets[52]. By identifying critical variables for salinity predictions, our approach encourages further application of these methods to the problem of freshwater salinity, specifically to develop a deeper understanding of the relative contribution of considered drivers to salinity changes across space and time. Such applications could also expand and further explore the impact of road salt and mining, which did not show significant contributions to salinisation for our study sites, but are known to be important contributors in other regions[18,38,53].

Although the datasets and models used in this study are associated with uncertainties, the use of global hydrological and climate models have become essential tools for cross-regional to global assessments[54]. For example, the sector-specific water use data from PCR-GLOBWB that we use, has been applied extensively in global water resource assessment studies[55], and has been validated to reported values and observed timeseries on various scales and to independent assessments[56]. Using state-of-the-art models and datasets available at the global scale therefore enabled us to conduct systematic assessments across these multiple regions and temporal scales, combined with the large amounts of salinity observational data. However, to complement the large-scale results of this study, refined analyses of irrigation-specific drivers (e.g., considering local irrigation techniques and specific irrigation water sources, as well as crop rotation practices) could aid in assessing model uncertainties and add more detailed knowledge of local-regional effects and feedbacks on salinity level and changes.

In summary, our analysis of freshwater salinity changes and its drivers in river sub-basins around the world stresses the large-scale contribution of irrigation land- and water use (water withdrawals, return flows and irrigated area) in regions experiencing salinisation issues. Future irrigation water use is projected to increase, driven by higher food demands with a growing world population[57]. Irrigation water demands may also increase due to hydroclimatic changes, which are expected to bring more severe drought events and dryland expansion[58]. In light of these expected future changes, the results of this study clearly highlight the need for timely action on freshwater salinity management. The identified contribution of irrigation-specific drivers to freshwater salinisation highlights the need to include these human components in further water quality assessments and for enabling reliable model predictions. Our results offer directions for the development of salinity-specific monitoring programs and management efforts of freshwater resources in agricultural regions.

## Methods

**River salinity data selection and processing.** For the selection of case study regions, we used electrical conductivity (EC) data from our previously synthesised global salinity dataset[26], in combination with data from the GEMStat[59] and the Mekong River commission data portal[60]. We considered all river monitoring locations from this dataset with a minimum timeseries of 15 years of monthly EC data availability within the period 1980–2010. In total, our dataset contains 417,315 salinity observations, from which we computed the monthly averages. The time period of 1980–2010 was selected due to its span over a 30-year period, which enables long-term trend assessment. This specific period is consistent with most of the driver variable datasets (see Table 1). The data criterion of 15 years of monthly available data allow locations with some data gaps, which is a very common trait for water quality monitoring data, while still ensuring long timeseries data spanning over at least half of the 30 years full time period considered.

From the monitoring locations fulfilling the selection criteria (see Supplementary Fig. 1 for all locations), we select seven regional river basins for further analyses; Mississippi (North America), Ebro, Danube, Rhine (Europe), Orange (Africa), Mekong (Asia) and Murray-Darling (Australia) (Fig. 2 and Supplementary Fig. 2 for measurement distributions and time series lengths). Beyond having multiple monitoring locations fulfilling the selection criteria, these river basins were selected to span different hydroclimatic and geographical regions and different anthropogenic impacts. Individual sub-basins from each selected salinity monitoring location within these seven regional river basins were then derived, using the hydrology toolset in ArcMap. We used the 15 arcsec flow direction data from HydroSHEDS[61], from which flow accumulation zones were computed. To adjust for EC locations being slightly off the river network, we run the "Snap Pour Point" tool to adjust and re-locate the EC sampling points to cells of high accumulated flow. All individual sub-basins with an area of less than 1 km² were excluded, due to the hydrologically unrealistic appearance of these sub-basins. This resulted in a set of 401 sub-basins within seven regional river basins around the world (zoomed panels of Fig. 2). The final set of sub-basin shapefiles was then exported and further processed in R (detailed processing steps below and in Supplementary Section S2).

**Driver variables selection and processing.** We acquired and processed a total of 26 geographic, hydroclimatic and human (mainly agricultural-related), driver variables, either as time series (monthly or annual) over the 1980–2010 period, or as constant values (see specifications in Table 1). The considered drivers were selected based on their expected influence on surface water salinity levels from a physical perspective and availability of data for selected river basins globally. Many anthropogenic activities may drive salinity changes, from which agricultural practices often are considered as having potentially large impacts. For example, irrigation can drive salinisation through different mechanisms; either through poor irrigation practices, raising saline groundwater volumes due to increased irrigation groundwater withdrawals[24], or from increasing evapotranspiration over irrigated fields[62]. For comparison purposes, we also considered sectoral water use and return flow beyond the irrigation sector, as these may also impact freshwater salinisation, e.g., through polluted runoff from urban and industrial sectors[63]. Regarding hydroclimatic variables, it is well known that salinity may increase with evaporation, due to evapo-concentration effects[64] and could change with changing discharge levels, due to changing dilution effects. Thus, we considered discharge, temperature, precipitation, as well as actual and potential evapotranspiration to be relevant hydroclimatic variables to be included in the analyses. Geographic variables that we considered, include soil salinity, elevation and distance to the coast, due to their known potential to influence salinity levels. Full processing steps of each of the considered driver variable are described in the supplementary information (Supplementary Section S2).

**Spatio-temporal analyses**

*Long-term salinity levels and threshold exceedance.* Long-term annual average salinity levels from each selected river monitoring location were used to classify the derived sub-basins into three overall salinity impact classes: Low (i.e., <700 µS cm⁻¹), Moderate (700–1500 µS cm⁻¹) and High (i.e., >1500 µS cm⁻¹) salinity (Fig. 2). The classification levels are based on global international threshold values for irrigation water use[27]. According to the Food and Agriculture Organization (FAO)[27], Electrical conductivity (EC) values of 700 µS cm⁻¹ represents slight to moderate restriction for irrigation uses and values >3000 µS cm⁻¹ represents severe restrictions. Zaman et al.[27] and references therein, define salinity levels <750 µS cm⁻¹ as having no effects, values 750–1500 µS cm⁻¹ risk having detrimental effects on sensitive crops, and levels >1500 µS cm⁻¹ having adverse effects on many crops. We used these global salinity threshold levels as a basis for our classification, due to the large-scale focus of our study and to allow for comparisons across river basins in different regions. Also, thresholds for irrigation water use are one of the most stringent, compared to other water use sectors.

In addition to classifying sub-basin average salinity conditions, we also computed threshold frequency exceedance. For this, we considered all monthly salinity observations and estimated the fraction (%) of months to the total number of months that each sub-basin exceeds the salinity threshold value of 700 µS cm⁻¹.

*Significance between salinity levels and driver variables*. To investigate the relation between sub-basin salinity and driver levels, long-term average values for each of the 26 driver variables (as listed in Table 1) within each salinity impact class (Low; $N = 263$, Moderate; $N = 108$, High; $N = 31$) were calculated over each sub-basin. The values were then normalised by dividing each sub-basin specific driver value by the group-average values, for each driver variable, and plotted over each salinity impact class (Low, Moderate, High). Boxplot statistics (including the median, interquartile ranges, whiskers) of driver contributions across the three salinity impact classes were summarised over the full dataset ($N = 401$; Fig. 4), as well as over each of the seven regional river basins (Supplementary Fig. 4).

Furthermore, we perform a nonparametric Wilcoxon rank sum test to assess differences in the distributions of each driver among the salinity impact classes (Low-High, Low-Moderate, Moderate-High), considering significance at the 95% significance level, i.e., $p < 0.05$ (Supplementary Tables 1 and 2). This test is useful when samples are independent (unpaired), which is assumed reasonable for this study, since each salinity impact class contains sub-basins of different characteristics.

*Trend analyses.* The nonparametric Mann-Kendall (MK) test[34,35] and Sen's slope estimation method[36] were applied to each sub-basin's timeseries values, for both salinity and driver variables (where possible, see specifications in Table 1 and Supplementary Section S2), to assess their variability and change over the 1980–2010 period. The MK test does not assume any data distribution pattern and is particularly useful when data are limited or incomplete, which is common for water quality monitoring data. MK tests have been widely used in environmental time series data studies, to which we refer for further details[65]. We used the MK test to determine whether there was a positive or negative trend in salinity and each considered driver. The strength of the trend was assessed using Kendall's tau (ranging from −1 to 1), with significant trends identified at 95% confidence level ($p$ value < 0.05). The magnitude of each sub-basin's time series trend (i.e., average salinity changes in µS cm$^{-1}$ per year) was then evaluated by the Sen's slope estimate. Specifically, for each river location, available instantaneous raw data over the 1980–2010 period was used, from which the trend tests were computed. For driver variables, either monthly or annual averages were used, based on timeseries data availability (Table 1).

**Random Forest model approach**. Random Forest (RF) is a well-established machine-learning algorithm[29], which is being increasingly used to improve understanding of the key-factors in environmental and freshwater problems[66]. The two different types of RF models are (i) classification, which is used for categorical values, and (ii) regression, which is used for discrete values. Although RF models were developed originally for prediction purposes, they are now also being used for association studies, i.e., for evaluating the extent to which a predictor plays a role for the prediction accuracy of the model. For this purpose, several variable importance measures have been implemented in RF approaches, with the purpose of ranking the importance of each predictor variable[67].

In this study we used RF regression to investigate the extent to which the different considered hydroclimatic, geographic and human drivers play a role for the prediction of observed salinity levels. We developed two RF regression models, one including all sub-basins with statistically significant increasing salinity trends ($N = 128$) and one including all sub-basins with statistically significant decreasing salinity trends ($N = 96$). Long-term annual average salinity values were used as the dependent variable for each sub-basin and the set of driver variables (both their long-term sub-basin averages, as well as each Sen's slope trend value) were used as independent variables.

*Model training and tuning.* A RF model attempts to predict outcomes for unseen data, based on what it has learnt from the training data. We trained each RF model with 80% of the dataset, randomly and proportionally selected from the salinity dataset. The predictive accuracy of the model was estimated with the remaining 20% of the data. The main tuning parameters of RF models are the number of trees; ntree, in the forest and the number of predictors sampled for each tree; mtry. The ntree parameter can be set to any number, but has been shown to continue to increase the RF accuracy up to a certain point, for which increasing the number after that threshold makes no model performance improvement[68]. Tuning was conducted, by varying ntree between 500, 750, 1000, 1500, 2000, 3000, 5000, 7500 and mtry between 1 and the maximum number of included driver variables. We then applied the optimal settings in each of the two RF models (Model 1; ntree=5000, mtry=12, model 2; ntree=5000, mtry=2).

*Driver importance evaluation.* It has been shown that correlated variables can impact the capacity of the model in identifying the strongest predictors, since any of the correlated variables can be used as the predictor from the model perspective, with no concrete preference of one over the others[69,70]. However, once one predictor is selected, the variable importance measure of the others is significantly reduced. This can lead to interpretation issues that one variable is a strong predictor, while others in the same group are not, whereas actually they are very close in terms of their relationship with the response variable[37].

Since several of the drivers considered here are correlated (Supplementary Fig. 7), we used conditional permutation importance (CPI)[37] to obtain a more

reliable assessment of each predictor's importance for salinity predictions. The CPI method is used to correct for the bias of feature importance when variables are correlated, due to its conditional (or partial) perspective. This means that the impact of a predictor variable is evaluated, while also considering the impact of all other variables. The CPI method has been shown to increase the interpretability and stability of RF computations in terms of variable importance, and has been used in a range of research areas[37]. A full methodological description of the original as well as the updated CPI method is given in Debeer and Strobl[37].

Specifically, the CPI method was implemented after the tuning of the ntree and mtry parameters, using the "permimp" function of the party package. This function allows the user to specify a threshold value $s$ ($0 \leq s \leq 1$, with 0.95 as default), which is a parameter that can be modified to make the CPI less or more conditional (1 representing completely unconditional). As recommended by the authors[37], we used different threshold values around the default (0.8-1), which enables us to study the changes in the CPI pattern between the importance of a predictor according to a more partial and a more marginal perspective. Due to the relative robustness of the results between these thresholds (viz. always the same top 15 predictors) and the high correlation between many of the considered variables, we present final results using a threshold value of 0.85.

After the CPI method had been used to identify the top 15 predictors, we further used recursive feature elimination[71]. to find the minimal set of variables leading to the best predictive capacity of each model. This was done through recursively running each RF model, starting with the top two predictors from the CPI method, and then adding one more variable in each run, in order of importance score until all variables had been added. The two final RF models include their respectively set of variables leading to the best model performance (Supplementary Table 3).

*Model performance.* The predictive performance of each RF model was evaluated using multiple statistical model performance indicators. Firstly, we compared observed salinity values with the predictions, by the squared correlation coefficient ($R^2$). In addition, to offer more in-depth assessment of model accuracy, we also calculated the Root Mean Squared Error (RMSE), which measures the absolute model error (in the same unit as the salinity measurements; µS cm$^{-1}$), as well as the Mean Absolute Error (MAE) and normalised MAE (NMAE), to get an estimate of the error magnitude.

## Data availability
The salinity, driver variables and sub-basin shapefile data generated and used in this study have been deposited in the "Salinity and drivers (human, geographic, hydroclimatic) datasets for assessing freshwater salinisation in river basins around the world" database under accession code (https://doi.org/10.5281/zenodo.4704824)[72]. The major part of the raw salinity datasets analysed in this study are publicly available in the PANGAEA repository, at: https://doi.org/10.1594/PANGAEA.913939[26]. The rest of the raw data can be downloaded or requested from the GEMStat[59] and the Mekong River commission data portal[60].

## Code availability
Code to generate the main figures of this study and for conducting the Random Forest modelling are available at Zenodo (https://doi.org/10.5281/zenodo.4704824)[72].

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

## Acknowledgements

We gratefully acknowledge Rens van Beek's contribution to initial PCR-GLOB WB model discussions and data sharing. This research was supported by The Swedish Research Council Formas (Project No. 2018-00812). Open access funding was provided by Stockholm University.

## Author contributions

J.T and M.T.H.v.V developed the idea of the paper and J.T was main responsible for data synthesis and analyses, as well as writing the manuscript. E.H.S was responsible for providing model output data from the PCR-GLOBWB 2 model. J.T, M.T.H.v.V, M.F.P.B, G.O.E and E.H.S contributed to the interpretation of the results. All authors provided feedback on the analyses and contributed to writing the manuscript.

## Funding

## Competing interests

The authors declare no competing interests.
