## [Peer Review File · Nature Communications]

REVIEWER COMMENTS

Reviewer #1 (Remarks to the Author):

Review of Thorslund et al, "Common drivers of freshwater salinisation in river basins worldwide"
(Nature Comms)

Salinization is a globally-relevant phenomenon with wide ranging concerns for agriculture, biodiversity, and human health. The drivers of salinity have been discussed extensively in regional studies, but cross-regional empirical analysis has been lacking, particularly on the scale presented by this manuscript. Using ~400 sub-basins from 7 distinct major river basins, the authors demonstrate that salinity has been changing across a range of environments in five continents. Their analysis highlights the inter-basin correlation between irrigation and salinization as well as the correlation between non-irrigation return flows (among other drivers) with long-term decreasing salinity. The paper is timely and represents an important empirical advancement to observing and assessing drivers of changing river salinity across regions. The manuscript is well written and generally contains appropriate data sources and analyses (a couple of exceptions/concerns are noted below, including that the analysis presented in Fig 4 should be closely examined). Although my understanding of the RF / CPI analysis suggests that it is highly appropriate for this paper, my understanding of the method is limited and I have noted a couple of requests for clarification under Specific Comments.

Further, I would like to raise a particular issue that merits further consideration within the manuscript. The authors find an (unexpected) inverse relationship between increasing salinity and PET/P (as well as AET/P), and hypothesize that it could be explained by greater evapo-concentration rates from soil evaporation versus plant transpiration. However, the proposed resolution contradicts our understanding of the physical mechanisms that produce salinization, as I describe below, and I am unconvinced by the effort to corroborate the hypothesis through an analysis of LAI and vegetation height. At minimum, this contradictory finding should be explored in more detail. Below, I elaborate on my reasoning and point to specific places in the text that offer counter arguments to the proposed hypothesis.

As the authors note, the finding that salinity is generally higher in regions with low PET/P (i.e., low aridity) contrasts with our general understanding of salinization, which is that it occurs most commonly in arid and semi-arid regions that require more irrigation (see line 277-278, which essentially state the same). The authors hypothesize that (line 152-154) the "domination of transpiration over bare soil evaporation in the ET signal could explain these patterns. In contrast to soil evaporation, which would drive salinisation through evapo-concentration, more vegetation and transpiration would not increase salinity levels." However, the latter part of this statement is contradicted by the statement on line 278 that "all salts in the irrigation water are not absorbed by the crops" which would, of course, lead to concentration of salts with transpiration. In other words, our understanding of the physical mechanisms of salinization indicate that the hypothesis should be viewed with skepticism. To test this hypothesis, the authors examine the relationship between salinity and LAI and salinity and vegetation height, but their findings (which they use to support their hypothesis) could simply be an artifact of the relationship between vegetation and climate (e.g., we generally expect greater LAI and vegetation heights in wetter regions where plants are less concerned with water conservation and embolism). I suggest that the authors explore potential interactions between aridity and irrigation demands and/or more carefully defend their proposed resolution to their contradictory finding. Additionally, there are multiple statements that pertain to this portion of the paper that should be referenced, as I note in the specific comments below.

Specific comments

L48-51: Could provide additional citations of existing studies of inland salinization in Australia (Rengasamy, 2006) and California (Schoups et al., 2005).

L54: It was at first unclear to me that all stations were converted to a monthly unit of analysis until reading the methods section (i.e., L317-319). This could be more explicit on lines 54-56. Additionally, I don't see how 401 stations with monthly observations over 30 years could lead to 400,000 salinity observations (401 sub-basins * 12 months * 30 years = 144,000). Lastly, it seems a better unit of analysis would be the sub-basin level (e.g., the fraction of months each station exceeds a salinity threshold).

L83: I had to re-read to ensure "variability" meant spatial variability. Perhaps add "variability across sub-basins" or something to similar effect to ensure this is explicit.

L89: This line reports the percent of total observations that exceed 700 μS , but the unit of analysis is confusing because the number of observations changes across each station (this is particularly true in the Danube which has a high variance in terms of the number of observations per stations, as shown in Figure S2). In other words, the results would be biased by stations with more observations. It should be demonstrated that the potential for such bias is small (or change the unit of analysis to the station, as mentioned above).

L151-153: The statement that soil evaporation would concentrate salts more than plant transpiration should be explained in more detail and cited.

L159-161: The observed relationship between higher LAI and vegetation height in sub-basins with lower salinity could be interpreted in a number of ways, depending on the cause of LAI and salinity. For instance, higher LAI could be the outcome of increased irrigation that promotes vegetation growth. Also see comment below (Fig S4) about the relationship between LAI and aridity.

L174: The caption to figure 4 indicates that the values for each driver are normalized by the group-average values of the driver (to produce the y-axis "normalized driver levels"). This approach is sensible to me, but it suggests that the means of each group presented in the figure should cluster around a value of 1 — which is not the case for some of the drivers. The normalization process should either be clarified or updated so that the mean values are centered around 1. Ideally, updating the normalization would reduce the overall range of values, allowing for a linear y-axis scale instead of the current logarithmic one. There is another issue that some of the means are less than half of the medians, which cannot be possible when the data are bounded by 0 on the lower end — and more broadly, for log-distributed data the mean should be greater than the median. Lastly, showing the statistical significance in this figure would be helpful (e.g., for Medium and High groups using Low as a baseline) so the reader does not have to refer to the SI.

L216: I suggest explaining the meaning (including units) of the trend outcomes (e.g., L236) from the analysis to make results easier to interpret.

L246: Additional explanation of the Accuracy Importance Score would be helpful — e.g., how should the units be interpreted?

L256: The analysis was presented on a per-measurement basis, not a per-station basis — in other words, the only analysis that would support this statement about temporal variability of salinity at the station level is in Figure 3B — which includes only 3 stations. See above comment for L54 and suggestion to convert analysis to a station basis.

L261: It would be helpful to provide an example or two that illustrate the extent to which salinity can reduce crop yield.

L277-280: The process of salt concentration in soils should be described in further detail and reconciled with the hypothesis presented on L151.

L343-345: Again, evapo-concentration processes should be clarified and reconciled with the hypothesis on L151.

Figure S4: It's quite odd that LAI is strongly positively correlated with the ET ratio (AET / P) and the aridity index (PET / P). This is the opposite of what is expected: the relationship arises due to the fact that plants in arid climates need to conserve water (smaller leaves) whereas plants in wetter climates compete for energy (bigger leaves). In other words, deviations from this relationship should be treated with caution. For one example, see Palmer et al. (2010).

Palmer et al. (2010). "Towards a spatial understanding of water use of several land-cover classes: an examination of relationships amongst pre-dawn leaf water potential, vegetation water use, aridity and MODIS LAI". *ECOHYDROLOGY*, 3(1), 1-10.

Rengasamy, P. (2006). World salinization with emphasis on Australia. *Journal of experimental botany*, 57(5), 1017-1023.

Schoups, Gerrit, et al. "Sustainability of irrigated agriculture in the San Joaquin Valley, California." *Proceedings of the National Academy of Sciences* 102.43 (2005): 15352-15356.

Reviewer #2 (Remarks to the Author):

This is a timely and relevant paper examining the salinity trends and some of their main causes for 401 sub-basins in 6 different regions of the world. The authors made a huge data compilation and analysis effort. The study covers wide temporal and spatial scales, and provides new information that can be very useful to manage freshwater salinisation around the world. The paper is well written and the statistical analyses are generally robust. Thus, I think that it has a great potential to be published in *Nature Communications* and to attract international attention. However, I have some comments and concerns:

I am afraid that the results of the study might be biased. The authors found that agriculture is the main driver of salinisation. However this is very likely influenced by the fact that most explanatory variables (Table 1) are related with agriculture and other important drivers of salinisation such as road salt (1, 2) and resource extraction (3-5) are not considered. These drivers can and should be incorporated into the study if a proper evaluation of the relative importance of each potential driver of salinisation is to be performed. The information on road traffic and mines is probably available for all catchments or can be provided by regional/national agencies. Alternatively, the authors should state more clearly that this paper focuses almost exclusively on agriculture and farming, and the title of the paper should also reflect this.

Also most of the information regarding drivers is taken from global data-sets and models that might not be very accurate. This is an important limitation that needs to be acknowledged and that could be addressed by using regionally available data.

The paper does not separate natural salinity (e.g. occurring in "Los Monegros" desert in the Ebro catchment and probably in some parts of the Mississippi) from anthropogenic salinisation (i.e. freshwater salinisation). This seems wrong when the title of the paper is "Common drivers of freshwater salinisation in river basins worldwide". If the aim of the paper is to assess the drivers of freshwater salinisation, the regions where high salinity occurs naturally should be excluded. This is important because natural saline streams have a high conservation value even if they don't provide water for irrigation, and they need to be preserved (6, 7).

Freshwater salinisation is commonly assessed from the point of view of aquatic biodiversity and ecosystem functioning. However, here the authors focus on the human use of water and use salinity classes that refer to irrigation water. A conductivity of 0.7 mS/cm, which falls within the "low salinity class", can be enough to have a significant impact on aquatic biodiversity (8, 9) and it is well above the official recommendations for the protection of aquatic life in Canada, Europe and the US. I think that this point deserves further discussion (although it is mentioned in lines 268-270).

Conductivity is a continuous variable. I don't fully understand why Wilcoxon tests between salinity classes are used to statistically assess the different contribution of each driver. Why not building linear mixed models (with the sub-basin identity as a random factor) or GAMs to assess the relationship between conductivity and each driver? This is partially addressed with Random Forests in the manuscript, but I still think that building models without categorising the sites could help to understand the relationship between the drivers and the conductivity values observed.

I wonder if the salinity trends could be broken into seasonal trends, with the salinisation trend (i.e. decreasing or increasing) depending on the season due to climatic factors and irrigation schemes.

Abstract

There are some papers analysing the causes of salinisation at regional scales (10–14). Thus, I am not sure if stating that "its drivers across regions are largely unknown", or it is more accurate to say that the knowledge is restricted to some regions in Australia, Europe and the US.

What do you mean by "consistent effects of higher contribution"? This is rather vague; you could be more specific.

Introduction

L26-27 This is a very general statement. I don't think it is needed.

L49 The models built by Le et al. (12) for Germany and Olson (13) for the US, indirectly provide information on the drivers of salinisation (i.e. more robust predictor variables have a stronger contribution to salinisation) that could be taken into account.

L154 In fact more vegetation could mean less salt in the water because the salt uptake by the roots.

L161 This makes sense. In naturally saline streams the vegetation is usually dominated by shrubs (7).

Figure 1: I miss road salt application and resource extraction. Also, naturally saline groundwaters can be a significant driver of salinization in some regions (15). Finally, I am not sure how dams act as a driver of salinization; this might need to be explained.

Figure 4: are all the boxplots showing statistically significant differences between classes? For some of them I see very weak differences.

Miguel Cañedo-Argüelles

References

1. W. D. Hintz, R. A. Relyea, A review of the species, community, and ecosystem impacts of road salt salinisation in fresh waters. *Freshw. Biol.* 64, 1081–1097 (2019).
2. S. S. Kaushal et al., Increased salinization of fresh water in the northeastern United States. *Proc. Natl. Acad. Sci.* 38, 13517–13520 (2005).
3. J. Bäche, E. Coring, Biological effects of anthropogenic salt-load on the aquatic Fauna: A synthesis of 17 years of biological survey on the rivers Werra and Weser. *Limnol. - Ecol. Manag. Int. Waters.* 41, 125–133 (2011).

4. M. A. Palmer et al., Mountaintop mining consequences. *Science* (80-.). 327, 148–149 (2010).
5. R. D. Vidic, S. L. Brantley, J. M. Vandebossche, D. Yoxtheimer, J. D. Abad, Impact of shale gas development on regional water quality. *Science*. 340, 1235009 (2013).
6. C. Gutiérrez-Cánovas et al., Do all roads lead to Rome? Exploring community trajectories in response to anthropogenic salinization and dilution of rivers. *Philos. Trans. R. Soc. B Biol. Sci.* 374 (2019) (available at <http://rstb.royalsocietypublishing.org/content/374/1764/20180009.abstract>).
7. A. Millán et al., Mediterranean saline streams in southeast Spain: What do we know? *J. Arid Environ.* 75, 1352–1359 (2011).
8. M. Cañedo Argüelles et al., Salinisation of rivers: an urgent ecological issue. *Environ. Pollut.* 173, 157–167 (2013).
9. Hart, et al., A review of salt sensitivity of Australian freshwater biota. *Hydrobiologia*. 210, 105–144 (1991).
10. S. S. Kaushal et al., Freshwater salinization syndrome on a continental scale. *Proc. Natl. Acad. Sci.* 115, E574–E583 (2018).
11. E. Estévez, T. Rodríguez-Castillo, M. G.-F. Alexia, M. Cañedo-Argüelles, J. Barquín, Drivers of spatio-temporal patterns of salinity in Spanish rivers: a nationwide assessment. *Philos. Trans. R. Soc. B Biol. Sci.* 374, 20180022 (2019).
12. T. D. H. Le et al., Predicting current and future background ion concentrations in German surface water under climate change. *Philos. Trans. R. Soc. B Biol. Sci.* 374, 20180004 (2019).
13. J. R. Olson, Predicting combined effects of land use and climate change on river and stream salinity. *Philos. Trans. R. Soc. B Biol. Sci.* 374, 20180005 (2019).
14. G. B. Allison et al., Land clearance and river salinisation in the western Murray Basin, Australia. *J. Hydrol.* 119, 1–20 (1990).
15. C. Li, X. Gao, S. Li, J. Bundschuh, A review of the distribution, sources, genesis, and environmental concerns of salinity in groundwater. *Environ. Sci. Pollut. Res.*, 1–18 (2020).

Manuscript Number: NCOMMS-20-50715

Manuscript Title: Common drivers of freshwater salinisation in river basins worldwide

Corresponding Author: Dr Josefin Thorslund

We would like to thank both reviewers, for their time and efforts in providing constructive feedback on our submitted manuscript. We believe that we could address and answer all questions raised through the review and that the review comments substantially helped to improve our manuscript. Please find below our detailed, point-by-point response to each of the comments of reviewer 1 and 2. We hope that with these revisions and improvements our manuscript can be accepted for publication in Nature Communications.

Yours sincerely,

Josefin Thorslund

Responses to Reviewer #1 comments

Reviewer #1 (Remarks to the Author):

Review of Thorslund et al, "Common drivers of freshwater salinisation in river basins worldwide" (Nature Comms)

1) Salinization is a globally-relevant phenomenon with wide ranging concerns for agriculture, biodiversity, and human health. The drivers of salinity have been discussed extensively in regional studies, but cross-regional empirical analysis has been lacking, particularly on the scale presented by this manuscript. Using ~400 sub-basins from 7 distinct major river basins, the authors demonstrate that salinity has been changing across a range of environments in five continents. Their analysis highlights the inter-basin correlation between irrigation and salinization as well as the correlation between non-irrigation return flows (among other drivers) with long-term decreasing salinity. The paper is timely and represents an important empirical advancement to observing and assessing drivers of changing river salinity across regions. The manuscript is well written and generally contains appropriate data sources and analyses (a couple of exceptions/concerns are noted below, including that the analysis presented in Fig 4 should be closely examined). Although my understanding of the RF / CPI analysis suggests that it is highly appropriate for this paper, my understanding of the method is limited and I have noted a couple of requests for clarification under Specific Comments.

→ *Thank you for these introductory comments and for judging our paper as an important contribution to this field. We have addressed your comments related to the RF / CPI analysis under Specific Comments below (see question 10-11). We hope that our revisions and additional clarifications will make these used methods more understandable to the general reader.*

2) Further, I would like to raise a particular issue that merits further consideration within the

manuscript. The authors find an (unexpected) inverse relationship between increasing salinity and PET/P (as well as AET/P), and hypothesize that it could be explained by greater evapo-concentration rates from soil evaporation versus plant transpiration. However, the proposed resolution contradicts our understanding of the physical mechanisms that produce salinization, as I describe below, and I am unconvinced by the effort to corroborate the hypothesis through an analysis of LAI and vegetation height. At minimum, this contradictory finding should be explored in more detail. Below, I elaborate on my reasoning and point to specific places in the text that offer counter arguments to the proposed hypothesis. As the authors note, the finding that salinity is generally higher in regions with low PET/P (i.e., low aridity) contrasts with our general understanding of salinization, which is that it occurs most commonly in arid and semi-arid regions that require more irrigation (see line 277-278, which essentially state the same). The authors hypothesize that (line 152-154) the "domination of transpiration over bare soil evaporation in the ET signal could explain these patterns. In contrast to soil evaporation, which would drive salinisation through evapo-concentration, more vegetation and transpiration would not increase salinity levels." However, the latter part of this statement is contradicted by the statement on line 278 that "all salts in the irrigation water are not absorbed by the crops" which would, of course, lead to concentration of salts with transpiration. In other words, our understanding of the physical mechanisms of salinization indicate that the hypothesis should be viewed with skepticism. To test this hypothesis, the authors examine the relationship between salinity and LAI and salinity and vegetation height, but their findings (which they use to support their hypothesis) could simply be an artifact of the relationship between vegetation and climate (e.g., we generally expect greater LAI and vegetation heights in wetter regions where plants are less concerned with water conservation and embolism). I suggest that the authors explore potential interactions between aridity and irrigation demands and/or more carefully defend their proposed resolution to their contradictory finding. Additionally, there are multiple statements that pertain to this portion of the paper that should be referenced, as I note in the specific comments below.

→ *Thank you for this explanatory and constructive comment. We agree that this contra-intuitive finding of higher salinity in lower aridity regions needs further consideration. We also agree that our originally proposed explanation of vegetation effects on salinisation in lower aridity regions may not be robust enough. The proposed relationship between salinity and vegetation parameters, as mentioned here, can indeed be an artefact between vegetation and climate (maybe even temperature driven) and we have therefore excluded this as a plausible explanation from the manuscript and associated analyses. In line with your suggestion, we have now further explored relations between aridity and irrigation. We explored these relations by data analysis of the long-term distribution of; irrigated area, irrigation consumption and irrigation water withdrawals, grouped over different aridity levels (estimated from the aridity index; PET/P) across all studied sub-basins. We find that all these **irrigation specific parameters are elevated in regions of lower aridity** (lower aridity index; PET/P), compared to in higher aridity regions. These supporting analyses are now presented in the new Supplementary Figure 5 (see also below). The observed higher irrigation activities in lower aridity regions are thus in line with our previous results between salinity classes and drivers; showing higher salinity levels with increasing irrigation activities, as well as higher salinity in regions with lower aridity (Fig. 4 of the main manuscript).*

Thus, higher irrigation water-use and activities (irrigation withdrawals and consumption, irrigated area) drive salinisation. This also explains the inverse relationship between salinity levels and aridity, since irrigation area and water use are elevated in less arid regions (new analyses in Supplementary Fig. 5).

Supplementary Figure 5. Relationships between sub-basin irrigation specific variables and aridity. Boxplots showing the distribution of sub-basin irrigated area (*Norm. Irr. area*) irrigation water use (irrigation water withdrawals; *Irr. ww* and irrigation consumption; *Irr. cons*), grouped over sub-basin aridity index (PET/P). The sub-basin aridity index classes are divided into three groups of similar sub-basin group sizes (N) and range from lower (<0.5) to higher (0.5-1 and >1) aridity, classified from long-term annual average values over the 1980-2010 period. Boxplot statistics include the median (vertical black lines), interquartile range (IQR: 25th percentile; Q1 and 75th percentile; Q3) and whiskers (confidence interval of $Q1 - 1.5 \times IQR$ to $Q3 + 1.5 \times IQR$). The stars (*) over the boxplots indicate where there are statistically significant differences in the irrigation related driver levels between low-high aridity groups (for <0.5 and > 1 groups; statistics based on Wilcoxon rank sum tests with a significance difference identified the 95% significance level, $p < 0.05$).

A likely explanation to why we observe lower irrigation water use with increasing aridity, could be due to **water limitations in the more arid regions** (172 of our investigated sub-basins occur in dry sub-humid to arid regions). While crop irrigation water demands are expected to be higher in more arid catchments, it is also more likely that the demand cannot be met due to a lack of available water resources (i.e., actual irrigation withdrawals and consumption is thus lower). Another possible explanation to the inverse relationship between irrigation and aridity could also be due to **irrigation-climate interactions**. Recent studies have shown that irrigation can lead to increasing soil moisture and relative humidity, which in turn, reduces atmospheric aridity (e.g., Ambika and Mishra, 2020 and references therein). Due to the land-atmospheric coupling processes

described above, irrigated regions across the world can therefore cause local/regional atmospheric cooling, which may reduce aridity (e.g. Thiery et al., 2017, Puma & Cook 2010). Such effects could then also contribute to why we observe relationships between lower aridity and increasing irrigation, which are in line with our findings of higher salinity in less arid regions.

Below are specific text revisions from the manuscript where these changes have been implemented and discussed, as well as the updated Fig. 4.

L194-207; "Surprisingly, for the hydroclimatic drivers, although we observe significant differences in evaporative ratios (PET/P, AET/P) and temperature (T) between salinity classes, the contribution of all these drivers decrease with increasing salinity levels (Fig. 4; direction across from green to orange boxplots). Such changes are opposite to expectations based on evapo-concentration effects (i.e. increasing salinity with increasing evaporation). It is also generally expected that the impact of irrigation in high salinity regions would correspond to more arid regions, which require more irrigation. To further investigate this unexpected correlation between salinity and aridity, we explore relations between aridity and irrigation within each sub-basin, including irrigated area, irrigation consumption and irrigation water withdrawals. We find that all these irrigation-specific parameters have a higher contribution in sub-basins of lower aridity (lower aridity index; PET/P), compared to higher aridity regions (Supplementary Figure 5). These results support our finding of the contribution of irrigation drivers in high salinity sub-basins, and explain the inverse relation between salinity levels and aridity, since irrigation area and water use are elevated in less arid regions. A likely explanation to why we observe lower irrigation water use with increasing aridity could be due to water limitations in the more arid regions (172 of our investigated sub-basins occur in dry sub-humid to arid regions). While crop irrigation water demands are expected to be higher in more arid catchments, it is also more likely that the demand cannot be met due to a lack of available water resources (i.e., actual irrigation withdrawals and consumption is thus lower). Another possible explanation to the inverse relationship between irrigation and aridity could be due to irrigation-climate interactions. Recent studies have shown that irrigation can lead to increasing soil moisture and relative humidity, which in turn, reduces atmospheric aridity³⁴. Due to the land-atmospheric coupling processes described above, irrigated regions across the world can therefore cause local/regional atmospheric cooling, which may reduce aridity³⁵. Such effects could also contribute to observed relationships between lower aridity and increasing irrigation, which are in line with our findings of higher salinity in less arid regions."

Figure 4. Contributions of hydroclimatic, geographic and human drivers across sub-basin salinity impact classes. Distribution of driver levels across sub-basin salinity impact classes, for a selection of 17 out of the total 26 considered driver variables. The selection includes all driver categories (as listed in Table 1), but where multiple variables within the same category exists (e.g. for soil salinity), only one variable was included. The salinity impact classes are based on groups of sub-basins with Low ($EC < 700 \mu S cm^{-1}$; green boxplots), Moderate ($EC = 700 - 1500 \mu S cm^{-1}$; orange boxplots) and High ($EC > 1500 \mu S cm^{-1}$; red boxplots) salinity levels, classified from long-term annual average values (as illustrated in Fig. 1). The selected drivers are plotted along groups of (i) hydroclimatic, (ii) geographic and (iii) human (agricultural-related) drivers on the x-axis, and their normalized levels on the y-axis. The values were normalized by dividing each sub-basin specific driver value by the group-average value, for each driver and then grouped over each salinity impact class. Full driver names and units are given in Table 1. Boxplot statistics include the median (vertical black lines), interquartile range (IQR: 25th percentile; Q1 and 75th percentile; Q3) and whiskers (confidence interval of $Q1 - 1.5 \cdot IQR$ to $Q3 + 1.5 \cdot IQR$). The stars (*) over the boxplots indicate where there are statistically significant differences in the driver contributions between the Low-High salinity impact classes (all summary statistics are included in Supplementary Table 1).

Specific comments

3) L48-51: Could provide additional citations of existing studies of inland salinization in Australia (Rengasamy, 2006) and California (Schoups et al., 2005).

→ *Both these references have now been included in the manuscript;*

L53-55: "In addition to hydroclimatic drivers¹⁷, relevant human drivers to inland freshwater salinisation include road salt application¹⁸, mining¹⁹ and agriculture²⁰ (Schoups et al., 2005)."

L57-62: "With regards to agricultural drivers, there are several studies at local and regional scales^{22,23} (Rengasamy, 2006), including raising saline groundwater due to irrigation withdrawals²⁴, or spreading of salt-containing fertilisers to the groundwater and/or surface water system²⁵. However, there is a lack of systematic assessments of the impact of agricultural activities on surface water salinisation at cross-regional to global scales.

Full references:

Rengasamy, P. (2006). World salinization with emphasis on Australia. *Journal of experimental botany*, 57, 1017-1023.

Schoups, Gerrit, et al. (2005). Sustainability of irrigated agriculture in the San Joaquin Valley, California. *Proceedings of the National Academy of Sciences*, 102, 43,15352-15356.

4) L54: It was at first unclear to me that all stations were converted to a monthly unit of analysis until reading the methods section (i.e., L317-319). This could be more explicit on lines 54-56. Additionally, I don't see how 401 stations with monthly observations over 30 years could lead to 400,000 salinity observations (401 sub-basins * 12 months * 30 years = 144,000). Lastly, it seems a better unit of analysis would be the sub-basin level (e.g., the fraction of months each station exceeds a salinity threshold).

→ *We computed monthly mean values for each station, from the 400,000 raw salinity measurement data available (some stations had daily instantaneous measurements, some only had monthly measurements). We agree that the information about the number of measurements and the monthly computations could be clarified and we have now made this more explicit (see text changes made below). Further, we have now updated the analyses of Fig. 3 to the sub-basin level (see updated Figure 3 and SM Figure 3 below). Fig. 3 now shows (a) the distribution of all sub-basin exceedance levels (i.e. the % of months each sub-basin station exceeds the salinity threshold of $700 \mu\text{S cm}^{-1}$), at a selection of three regional river basins, and (b) timeseries of salinity exceedance at individual sub-basin stations, with one sub-basin example from the Low salinity impact class (selected from the longest station timeseries available) from each of the regions in (a). The boxplot in (a) has been separated to show the exceedance level of all sub-basins within the Low salinity impact class (i.e. sub-basins with long term annual average salinity values below 700*

$\mu\text{S cm}^{-1}$), compared to all sub-basins within the Moderate-High salinity impact class (i.e. long-term salinity levels above this threshold; in accordance with Fig. 2), to highlight that sub-basins with long-term levels below this threshold still temporally exceed $700 \mu\text{S cm}^{-1}$ to various degrees. In addition to these three examples in the main manuscript, the supplementary materials (new Supplementary Figure 3, also included below) now also shows the distribution of sub-basin exceedance frequency levels for all, as well as for each of the seven regional basins. We have updated the abstract and result text with regards to these analysis changes;

L70-75 (with regards to monthly analysis from raw data): "Within these regions, we delineate the sub-basins from all river monitoring locations with a minimum of 15 years of monthly salinity data within the period 1980-2010 (Supplementary Figure 1-2). We compute monthly mean salinity levels of stations from over 400,000 salinity observations (electrical conductivity; EC), synthesised from a new open-access global database³⁰."

L111-132 (for sub-basin analysis level and results): "Compared to these long-term average levels, we also analyse sub-basin exceedance frequency of salinity values above the $700 \mu\text{S cm}^{-1}$ irrigation threshold, both within and across regions (i.e. the fraction of months compared to total that each sub-basin exceeds the salinity threshold). We separate the effects to show the exceedance level of all sub-basins within the "low" salinity impact class (i.e. sub-basins with long term annual average salinity values below $700 \mu\text{S cm}^{-1}$), compared to all sub-basins within the "moderate-high" salinity impact class (i.e. long-term salinity levels above this threshold). Examples from the Mississippi, Ebro and Orange basins highlight that even when long-term salinity levels are below the $700 \mu\text{S cm}^{-1}$ threshold (i.e. "low" salinity class), sub-basin exceedance still occur on shorter timescales (blue boxplots of Fig. 3a) and there can be large intra-annual variability in salinity levels (Fig. 3b). We see similar results considering all regions, with an average sub-basin exceedance level of 33 % (Supplementary Figure 3a), but with highly variable results (sub-basin exceedance levels of between 0-58 %; Supplementary Figure 3b). Such intra-annual threshold exceedance can have important implications for water use management, as these water resources may not always be of suitable salinity level for irrigation water use purposes."

Figure 3. Distribution and timeseries of sub-basin salinity threshold exceedance. Panel (a) shows three examples of the distribution of sub-basin salinity exceedance levels (% of months exceeding the threshold $700 \mu\text{S cm}^{-1}$ compared to total number of monthly measurements for each sub-basin) within the Mississippi, Ebro and Orange regional river basins, grouped over sub-basins within the *Low* salinity impact class (i.e. sub-basins with long term annual average salinity values below $700 \mu\text{S cm}^{-1}$), and sub-basins within the *Moderate-High* salinity impact class (i.e. long-term salinity levels above this threshold). Panel (b) shows timeseries of one individual sub-basin station from the *Low* salinity impact class (selected from the longest timeseries available), from each river basin example in (a), with salinity levels exceeding the threshold highlighted in red.

Supplementary Figure 3. Distribution of sub-basin salinity threshold exceedance level within each regional basin. Boxplot showing exceedance level (% of months compared to total that each sub-basin exceeds the salinity threshold) within each regional river basin, as well as for all sub-basins together (panel a), and (panel b) grouped over sub-basins within the *Low* salinity impact class (i.e. sub-basins with long term annual average salinity values below $700 \mu\text{S cm}^{-1}$), and sub-basins within the *Moderate-High* salinity impact class (i.e. long-term salinity levels above this threshold).

5) L83: I had to re-read to ensure “variability” meant spatial variability. Perhaps add “variability across sub-basins” or something to similar effect to ensure this is explicit.

→ *This has now been changed to (L107-110); “The overall largest variability across sub-basins is seen in the Mississippi, Ebro and Murray-Darling river basins, while the Mekong and the Danube river basins show the overall lowest variability across sub-basins (while dominated by the Low salinity impact class).”*

6) L89: This line reports the percent of total observations that exceed 700 μS , but the unit of analysis is confusing because the number of observations changes across each station (this is particularly true in the Danube which has a high variance in terms of the number of observations per stations, as shown in Figure S2). In other words, the results would be biased by stations with more observations. It should be demonstrated that the potential for such bias is small (or change the unit of analysis to the station, as mentioned above).

→ *We have now changed the unit of analysis to the sub-basin level as mentioned in our response to question 4 above and updated the text to highlighting this, see for example:*

L111-126: “Compared to these long-term average levels, we also analyse sub-basin exceedance frequency of salinity values above the 700 $\mu\text{S cm}^{-1}$ irrigation threshold, both within and across regions (i.e. the fraction of months compared to total that each sub-basin exceeds the salinity threshold). We separate the effects to show the exceedance level of all sub-basins within the Low salinity impact class (i.e. sub-basins with long term annual average salinity values below 700 $\mu\text{S cm}^{-1}$), compared to all sub-basins within the Moderate-High salinity impact class (i.e. long-term salinity levels above this threshold). Examples from the Mississippi, Ebro and Orange basins highlight that even when long-term salinity levels are below the 700 $\mu\text{S cm}^{-1}$ threshold (i.e. Low salinity impact class), sub-basin exceedance still occur on shorter timescales (blue boxplots of Fig. 3a) and there can be large intra-annual variability in salinity levels (Fig. 3b).”

7) L151-153: The statement that soil evaporation would concentrate salts more than plant transpiration should be explained in more detail and cited.

→ *As explained in our response to comment/question 2 above, we have now updated the explanatory analyses from the original explanation of vegetation and associated transpiration over bare soil evaporation. We now focus on interactions between aridity and irrigation demands, as suggested by this reviewer. The statement on soil evaporation has thus been removed and this section now reads;*

L179-186: “To further investigate this unexpected correlation between salinity and aridity, we explore relations between aridity and irrigation within each sub-basin, including irrigated area, irrigation consumption and irrigation water withdrawals. We find that all these irrigation-specific parameters have a higher contribution in sub-basins of lower aridity (lower aridity index; PET/P), compared to higher aridity regions (Supplementary Figure 5). These results support our finding of the contribution of irrigation drivers in high salinity sub-basins, and explain the inverse relation

between salinity levels and aridity, since irrigation area and water use are elevated in less arid regions.”

8) L159-161: The observed relationship between higher LAI and vegetation height in sub-basins with lower salinity could be interpreted in a number of ways, depending on the cause of LAI and salinity. For instance, higher LAI could be the outcome of increased irrigation that promotes vegetation growth. Also see comment below (Fig S4) about the relationship between LAI and aridity.

→ As clarified above (please see our full response to comment nr 2), we have now updated the analyses and removed the parts relating to the vegetation aspects, as we agree this hypothesis did not properly explain the observed relationships between salinity and aridity. In line with your suggestion, we have now further explored relations between aridity and irrigation.

9) L174: The caption to figure 4 indicates that the values for each driver are normalized by the group-average values of the driver (to produce the y-axis “normalized driver levels”). This approach is sensible to me, but it suggests that the means of each group presented in the figure should cluster around a value of 1 — which is not the case for some of the drivers. The normalization process should either be clarified or updated so that the mean values are centered around 1. Ideally, updating the normalization would reduce the overall range of values, allowing for a linear y-axis scale instead of the current logarithmic one. There is another issue that some of the means are less than half of the medians, which cannot be possible when the data are bounded by 0 on the lower end — and more broadly, for log-distributed data the mean should be greater than the median. Lastly, showing the statistical significance in this figure would be helpful (e.g., for Medium and High groups using Low as a baseline) so the reader does not have to refer to the SI.

→ The normalization was computed as; individual sub-basin driver value, divided by the average driver value of the full dataset (i.e. all 401 sub-basins), for each driver variable. It is correct that the mean of each driver group clusters around 1 for the full dataset (i.e. ind. sub-basin value/all sub-basin values). However, we then group and plot these normalised values over each salinity impact class (Low, Moderate, High), for the possibility of investigating the effects of each driver level across different salinity classes. Therefore, the means of each salinity class group will not be 1 (since it is a subset of the full normalized group, which has a mean of 1). If we would normalise by each salinity impact class, then they would all be clustered around 1 and we could not study the effects of each driver across different salinity levels (i.e. it would remove any effects of different driver levels between classes if each class is normalised by itself). We have now clarified this in the figure caption (see updated figure 4 in our response to comment 2 above) and in the methods at;

L498-506: *“To investigate the relations between sub-basin salinity and driver levels, long-term average values for each of the 26 driver variables (as listed in Table 1) within each salinity class (Low; N=263, Moderate; N=108, High; N=31) were calculated over each sub-basin. The values were then normalised by dividing each sub-basin specific driver value by the group-average values, for each driver variable, and plotted over each salinity class (low, moderate, high). Boxplot*

statistics (including the median, interquartile ranges, whiskers) of driver contributions across the three salinity classes were summarized over the full dataset (N= 401; Fig. 4), as well as over each of the seven regional river basins (Supplementary Figure 4)."

Regarding the logarithmic scale chosen for plotting, this was done to have a clearer visual representation of the boxplots and to better see their spread between salinity groups, as the range of values are relatively large and therefore the difference between some driver levels (e.g. those that are clustered close to 0) would be hard to distinguish between the salinity groups. Due to the logarithmic scale there was however an error in the displayed mean values, which have now been corrected. Instead, as suggested, **we have now visualized** which drivers show significant different contributions between salinity groups – **by including a * above each boxplot where there is a statistically significant difference in mean driver levels** between low and high groups (in accordance with the results of SM Table 1). We hope that these modifications will make the results of Fig. 4 easier for the reader to interpret and without having to move to the SM to look this up statistics.

10) L216: I suggest explaining the meaning (including units) of the trend outcomes (e.g., L236) from the analysis to make results easier to interpret.

11) L246: Additional explanation of the Accuracy Importance Score would be helpful — e.g., how should the units be interpreted?

→ We here address both comments 10 and 11. The results of the RF and the CPI analysis provide new understanding of the relative importance of different variables in achieving the predictive capacity of the model. The CPI accuracy score is unitless and should be seen as a relative scale, rather than actual values (i.e. a higher score means that the variable is more important to include in the model than a variable with a lower score). We have now added some more explanations within the manuscript text to make the RF/CPI analyses and results easier to understand;

L266-270: "For sub-basins experiencing salinisation, our results clearly show that irrigated area (Norm. irr. area; Fig 6a), plays the most significant role in achieving prediction accuracy of the RF model, out of the 26 included driver variables. This is identified by its higher CPI accuracy importance score compared to the other variables (unitless; a higher score means a higher relative contribution compared to other variables)."

L303-320: "To further understand the absolute contributions of the different drivers between sub-basins of increasing and decreasing salinity levels, a more detailed analysis of actual driver levels and/or trends should be considered. For example, irrigation return flows (Irr. rf) emerge among the top 15 predictors for both increasing and decreasing salinity trends (Fig. 6a-b). This means that this variable is important to consider when predicting salinity levels, regardless of the underlying salinity trend. However, actual driver impact level must be related to further analyses, for instance combined with correlations assessment (Fig. 4) or other statistical analyses. The results of the RF and the CPI analyses are valuable tools for moving away from "black-box" type predictions, by identifying variables that are critical for successful salinity predictions. Using these methods in

regions experiencing contrasting salinity trends is an important first step for improving our understanding of relations between salinity levels, trends and drivers."

12) L256: The analysis was presented on a per-measurement basis, not a per-station basis — in other words, the only analysis that would support this statement about temporal variability of salinity at the station level is in Figure 3B — which includes only 3 stations. See above comment for L54 and suggestion to convert analysis to a station basis.

→ *As mentioned above (please see our detailed response to comment nr 4), we now have temporal exceedance levels (% of total months that each sub-basin exceeds the salinity threshold) from all sub-basins included in the analyses. In the main manuscript, we now show the distribution of all sub-basins exceedance levels within three regions (Fig. 3a), and in the Supplementary materials we show results for all regions and all associated sub-basins (new Supplementary Fig. 3).*

13) L261: It would be helpful to provide an example or two that illustrate the extent to which salinity can reduce crop yield.

→ *We have now included an example of observed yield reductions to sensitive crops from irrigating with saline water in the discussion;*

L336-342; "Irrigating with saline water could severely affect crop yields, due to long-term soil salinisation build-up, waterlogging and reduced water availability for crops when salinity increases⁴³. Yield reductions of between 10-25 % have been shown to affect saline sensitive crops (e.g. beans, carrots and lettuce) when irrigating with saline water of around 1000 $\mu\text{S cm}^{-1}$ (i.e. salinity levels within the moderate and high salinity classes of this study)⁴³. The risks of higher yield reductions also increase over time, due to soil salinity build-up⁴⁴."

14) L277-280: The process of salt concentration in soils should be described in further detail and reconciled with the hypothesis presented on L151.

→ *We have now updated the analysis and associated text to explaining the inverse relationship between salinity and aridity by interactions between aridity and irrigation demands, and excluded the previous hypothesis of salt concentration in soils due to vegetation effects (please see our full response to your comment 2 above).*

15) L343-345: Again, evapo-concentration processes should be clarified and reconciled with the hypothesis on L151.

→ *See our response to question 14 above. However, with regards to mentioning of evapo-concentration processes in the methods, we still think this is reasonable and valid process to bring up.*

16) Figure S4: It's quite odd that LAI is strongly positively correlated with the ET ratio (AET / P) and the aridity index (PET / P). This is the opposite of what is expected: the relationship arises due to the fact that plants in arid climates need to conserve water (smaller leaves) whereas plants in wetter climates compete for energy (bigger leaves). In other words, deviations from this relationship should be treated with caution. For one example, see Palmer et al. (2010).

→ *As mentioned in our detailed response (see our full response under comment 2), we have now removed the vegetation part and believe we can better explain the elevated salinity levels in lower aridity regions due to higher irrigation activities in less arid sub-basins, rather than with vegetation patterns.*

Palmer et al. (2010). "Towards a spatial understanding of water use of several land-cover classes: an examination of relationships amongst pre-dawn leaf water potential, vegetation water use, aridity and MODIS LAI". *ECOHYDROLOGY*, 3(1), 1-10.

Rengasamy, P. (2006). World salinization with emphasis on Australia. *Journal of experimental botany*, 57(5), 1017-1023.

Schoups, Gerrit, et al. "Sustainability of irrigated agriculture in the San Joaquin Valley, California." *Proceedings of the National Academy of Sciences* 102.43 (2005): 15352-15356.

New references included in the author response to reviewer #1:

Ambika, A. K. & Mishra, V. Substantial decline in atmospheric aridity due to irrigation in India. *Environ. Res. Lett.* 15, 124060 (2020).

Puma, M. J. & Cook, B. I. Effects of irrigation on global climate during the 20th century. *Journal of Geophysical Research: Atmospheres* 115, (2010).

Thiery, W. et al. Present-day irrigation mitigates heat extremes. *Journal of Geophysical Research: Atmospheres* 122, 1403–1422 (2017).

Reviewer #2 (Remarks to the Author):

This is a timely and relevant paper examining the salinity trends and some of their main causes for 401 sub-basins in 6 different regions of the world. The authors made a huge data compilation and analysis effort. The study covers wide temporal and spatial scales, and provides new information that can be very useful to manage freshwater salinisation around the world. The paper is well written and the statistical analyses are generally robust. Thus, I think that it has a great potential to be published in Nature Communications and to attract international attention. However, I have some comments and concerns:

1) I am afraid that the results of the study might be biased. The authors found that agriculture is the main driver of salinisation. However, this is very likely influenced by the fact that most explanatory variables (Table 1) are related with agriculture and other important drivers of salinisation such as road salt (1, 2) and resource extraction (3–5) are not considered. These drivers can and should be incorporated into the study if a proper evaluation of the relative importance of each potential driver of salinisation is to be performed. The information on road traffic and mines is probably available for all catchments or can be provided by regional/national agencies. Alternatively, the authors should state more clearly that this paper focuses almost exclusively on agriculture and farming, and the title of the paper should also reflect this.

→ *Dear Miguel Cañedo-Argüelles, thank you for these constructive comments and for highlighting the great potential of our study for publication in Nature Communications. Regarding the selection of drivers, in this study, we focus particularly on evaluating the impact of agricultural activities (including irrigation water use and return flows, fertilizer use) on salinity patterns, because of its argued relevance in smaller scale studies (e.g. Buvaneshwari et al., 2020, Barros et al., 2012, Kass et al. 2005), but lack of systematic assessments across large spatial (cross-regional to global) and temporal scales. However, we understand your point regarding risk of bias and we have now performed additional analyses with regards to impacts of mining and road salt, to test the robustness of our results.*

For mining, we used a new global gridded dataset of mining areas (Maus et al. 2020) from which we evaluated differences in mining area (total and normalized) across sub-basin salinity impact levels. In addition, we evaluated its significance for predicting salinity levels in regions of increasing salinisation, using the random forest approach. We did not find any significant relation between mining and salinity with regards to a higher contribution across higher salinity impact classes (see new Supplementary Figure 6 below). The mining variables were also ranked very low in the CPI analysis for predicting salinity levels in sub-basins experiencing salinisation (see new Supplementary Figure 8a below). We do still acknowledge that mining can be a significant contributor to salinisation issues in other regions, and we have included it in the discussion (see text below), but our revised analyses do not indicate any significant contributions of mining to the observed and predicted salinity levels within our study basins.

With regards to road salts, due to the lack of systematic datasets across regions and lack of relevance in many of our considered sub-basins (many of the regions considered are in climate

zones with snow-free winters, where road salt application is not present or limited) we estimate annual average application (in pounds) within available years of our study period (1992-2010) (Bock et al. 2018) for all sub-basins within the Mississippi river basin as a case study (N=167). Using the Mississippi region as a case study is motivated since road-salt usage is known to occur in this river basin region of the US (e.g. Kaushal et al. 2018, Kelly et al., 2012) and because it is a large proportion of the total sub-basin dataset of this study. Similar to mining, we did not find a significant contribution of road salt in sub-basins of elevated salinity levels (SM Fig. 6 below), nor as a top predictor variable within the RF and CPI analyses scores (SM Fig. 8b below).

The new analyses of road salt and mining have now been included in the manuscript in the new Supplementary Figure 6 and 8, and in the result and discussion section, as outlined below. Taken together, we believe these new driver analyses provide additional evidence to the robustness of our results regarding irrigation activities being the main drivers of elevated salinity and increasing salinisation. These additional drivers (i.e. mining and road salt) do not impact our conclusions. With regards to the main focus of the study, we do however agree that we should have been clearer about our focus of agricultural and irrigation-specific drivers and have also, as suggested, updated the title of the paper to better reflect this (new title: "Common irrigation drivers of freshwater salinisation in river basins worldwide"), as well as made updates throughout the whole manuscript text, as exemplified below. We hope that these modifications help to better reflect the scope and results of our paper.

Text modifications with regards to mining and road salt:

L223-235 (result): "Although our focus is on evaluating the impact of agricultural-related human drivers, we perform additional analyses to quantify possible impacts of mining and road salt use on salinity levels, to test the robustness of our results. We use a new global gridded dataset of mining areas³⁶, to evaluate the contribution of mining area (total and normalized) within all sub-basins, across salinity impact levels. For road salt, we estimate annual average application (in pounds) within available years of our study period (1992-2010)³⁷ for all sub-basins within the Mississippi river basin as a case study (N=167). We focus on this region, due to due to limitations of global datasets and the lack of relevance of road salt in many of our considered regions (located in climate zones with snow-free winters, where road salt application is not present/limited). We did not find any significant contributions of road salt or mining in sub-basins within the High or Moderate salinity impact class (Supplementary Figures 6)."

L295-302 (result): "In line with earlier analysis, we also included mining and road salt as predictor variables in the RF and CPI-analysis, to predict salinity levels in sub-basins experiencing salinisation. Mining showed a very low CPI accuracy importance score compared to the other variables, which indicates its lack of relevance for predicting salinity levels within the here studied regions (Supplementary Figure 8a,b). For the Mississippi case study example, road salt was more important than mining, but was still only the 14th highest ranked variable in this region, scoring lower than all irrigation-related parameter (Supplementary Figure 8a,b)."

L391-397 (discussion section): "By identifying critical variables for salinity predictions, our approach encourages further application of these methods to the problem of freshwater salinity, specifically to develop a deeper understanding of the relative contribution of considered drivers to salinity changes across space and time. Such applications could also expand and further explore the impact of road salt and mining, which did not show significant contributions to salinisation for our study sites, but are known to be important contributors in other regions^{18,52,42}."

Supplementary Figure 6. Contributions of mining and road salt drivers across sub-basin salinity impact levels.

Boxplots showing the distribution of sub-basin mining area (Norm. Mining area and Tot. Mining area)²⁵ and road salt application (in pounds; annual average application over the years 1992-2010)²⁶ grouped over sub-basin salinity impact classes. The mining area drivers are quantified for all sub-basins (N=401), whereas the road salt driver is quantified over the Mississippi sub-basins as a case study example (N=167). Boxplot statistics include the median (vertical black lines), interquartile range (IQR: 25th percentile; Q1 and 75th percentile; Q3) and whiskers (confidence interval of $Q1 - 1.5 \cdot IQR$ to $Q3 + 1.5 \cdot IQR$).

Supplementary Figure 8. Relative driver contributions, including mining and road salt drivers, for predicting salinity levels in sub-basins with increasing salinisation. Results from the CPI analysis when including mining (sub-basin normalized mining area and total mining area) and road (sub-basin total annual application) salt as predictor variables in the RF model to predict salinity levels in sub-basins experiencing salinisation. Bar charts show the relative importance of driver variables in (a) all sub-basins and (b) for sub-basins within the Mississippi river basin (a relevant case study for road salt application), using the conditional permutation importance method within the Random Forest approach. For illustration purposes, only the top 10 drivers, as well as the relative importance of mining and road salt are shown. The accuracy importance score (y-axis; unitless) represent the relative importance of each predictor variable in achieving the prediction capacity of the Random Forest model (note the different y-axes in panel a and b and that the absolute values of the scores should not be interpreted, only the relative scores between drivers).

Text modifications with regards to better highlighting the focus of the study:

L18-28 (abstract): *"Here, we assess inland freshwater salinity patterns and evaluate its interactions with irrigation water use, across seven regional river basins (401 river sub-basins) around the world, using long-term (1980-2010) salinity observations. While a limited number of sub-basins show persistent salinity problems, many sub-basins temporarily exceeded safe irrigation water-use thresholds and 57 % experience increasing salinisation trends. We further investigate the role of agricultural activities as drivers of salinisation and find common contributions of irrigation-specific activities (irrigation water withdrawals, return flows and irrigated area) in sub-basins of high salinity levels and increasing salinisation trends, compared to regions without salinity issues."*

L53-62 (introduction): *"In addition to hydroclimatic drivers¹⁷, relevant human drivers to inland freshwater salinisation include road salt application¹⁸, mining¹⁹ and agriculture²⁰. Among these human drivers, the impact of road salts on rising salinity levels has been relatively well-quantified and reviewed across multiple inland surface water bodies and regions^{18,21}. With regards to agricultural drivers, there are several studies at local and regional scales^{22,23}, including raising saline groundwater due to irrigation withdrawals²⁴, or spreading of salt-containing fertilisers to the groundwater and/or surface water system²⁵. However, there is a lack of systematic assessments of the impact of agricultural activities on surface water salinisation at cross-regional to global scales."*

L82-90 (introduction): *"We further acquire global-scale open data of agriculturally relevant variables for evaluating their contributions as drivers to observed salinity levels, as well as some additional hydroclimatic and geographic variables that are expected to influence salinity levels. Sub-basin averages of these 26 driver variables (Table 1 and Supplementary section S2) are calculated and their relationship across salinity levels and trends are assessed, using both statistical and machine learning (random forest (RF)) methods³³."*

L134-138 (results): *"To investigate the contributions of the selected set of driver variables to observed salinity levels, we quantify long-term (i) (agriculture-related) human (ii) hydroclimatic and (iii) geographic variables over each sub-basin, and compare their contributions across the different salinity impact classes (Low, Moderate, High)."*

L458-464 (methods): *"We acquired and processed a total of 26 geographic, hydroclimatic and human (mainly agricultural-related), driver variables, either as time series (monthly or annual) over the 1980-2010 period, or as constant values (see specifications in Table 1). The considered drivers were selected based on their expected influence on surface water salinity levels from a physical perspective and availability of data for selected river basins globally. Many anthropogenic activities may drive salinity changes, from which agricultural practices often are considered as having potentially large impacts."*

2) Also, most of the information regarding drivers is taken from global data-sets and models that might not be very accurate. This is an important limitation that needs to be acknowledged and that could be addressed by using regionally available data.

→ *For the spatial scope of this cross-regional study, covering a range of sub-basins in different continents, we wanted to synthesize and use freely available global-scale datasets. Using the same datasets for all sub-basins enables systematic cross-regional assessments, which may substantially reduce bias (or at least bring the same level of bias) across the quantified sub-basins, compared to using local/national agency datasets. It also facilitates re-usability and expansion of our findings for future research, which could expand to other or more regions/sub-basins, while still using the same datasets for the considered drivers. There is of course always a trade-off when working with large-scale data, both from observed datasets as well as from models. Using local/regional datasets also comes with uncertainties and limitations (e.g. different ways and/or scales of measuring/reporting water use data or soil salinity, different methods for estimating climate parameters such as rain water gauges or evapotranspiration), that may not necessarily bring more accurate results when compiling and comparing different datasets at these cross-regional scales. In addition, as indicated for most variable descriptions in the Supplementary Materials (S2), the datasets and models that we used have been validated through extensive cross-validation and error checking using a multitude of sources and statistical approaches, and have been commonly used in cross-regional to global assessments like our study (e.g. Tangdamrongsub et al. 2017).*

We agree that global data-sets and model results also have their limitations and uncertainties and we see the value in using measured data, which is why we also made the huge synthesis effort with the river salinity monitoring data from local and regional sources (Thorslund & van Vliet 2020). However, with regards to drivers, we feel that it was most feasible to use global datasets, due to lack of available driver information from local/regional sources for all 401 sub-basins. We have now further acknowledged the impacts of uncertainties associated with using global datasets and models in the discussion (see below for text-specific updates). We have also added validation information regarding the PCR-GLOBWB data in the Supplementary materials (S2) (see below) to acknowledge the impacts of uncertainties in global datasets used.

L398-411: "Although the datasets and models used in this study are associated with uncertainties, the use of global hydrological and climate models have become essential tools for cross-regional to global assessments⁵³. For example, the sector-specific water use data from PCR-GLOBWB that we use, has been applied extensively in global water resource assessment studies⁵⁴, and has been validated to reported values and observed timeseries on various scales and to independent assessments⁵⁵. Using state-of-the-art models and datasets available at the global scale therefore enabled us to conduct systematic assessments across these multiple regions and temporal scales, combined with the large amounts of salinity observational data. However, to complement the large-scale results of this study, refined analyses of irrigation-specific drivers (e.g. considering local irrigation techniques and specific irrigation water sources, as well as crop rotation practices) could aid in assessing model uncertainties and add more detailed knowledge of local-regional effects and feedbacks on salinity level and changes."

SM, section S2: "The water allocation data simulated by PCR-GLOBWB 2 has been validated to both reported values and to independent assessments on country, continent and global scales^{4,6}."

3) The paper does not separate natural salinity (e.g. occurring in "Los Monegros" desert in the Ebro catchment and probably in some parts of the Mississippi) from anthropogenic salinisation (i.e. freshwater salinisation). This seems wrong when the title of the paper is "Common drivers of freshwater salinisation in river basins worldwide". If the aim of the paper is to assess the drivers of freshwater salinisation, the regions where high salinity occurs naturally should be excluded. This is important because natural saline streams have a high conservation value even if they don't provide water for irrigation, and they need to be preserved (6, 7).

→ *It is correct that we do not explicitly separate or exclude naturally saline river basins within our dataset. However, if the fact that we see a strong signal of irrigation related drivers in basins with higher salinity would be impacted by a set of naturally saline basins – our results would rather indicate conservative estimates of the importance of irrigation as a driver of salinisation (i.e. possibly underestimating, not overestimating). This is because, if our dataset includes naturally saline sub-basins (i.e. where salinity levels would then not be correlated to high irrigation levels), this would cause an overall weaker correlation to irrigation levels across our set of high salinity sub-basins. Therefore, the potential inclusion of non-human affected saline basins does not affect the interpretation of our results. Nevertheless, also in response to your previous point (question 1), we have changed the title of the manuscript to emphasize that we focus on common irrigation drivers.*

4) Freshwater salinisation is commonly assessed from the point of view of aquatic biodiversity and ecosystem functioning. However, here the authors focus on the human use of water and use salinity classes that refer to irrigation water. A conductivity of 0.7 mS/cm, which falls within the "low salinity class", can be enough to have a significant impact on aquatic biodiversity (8, 9) and it is well above the official recommendations for the protection of aquatic life in Canada, Europe and the US. I think that this point deserves further discussion (although it is mentioned in lines 268-270).

→ *We agree that levels below our salinity thresholds can still bring impacts for aquatic biodiversity and have added some more discussion around its relevance (see below). However, we hope that with the clarifications and streamlining of the paper - to now better reflect the focus of the impacts for and of the agricultural sector on freshwater salinity levels – that this is now more in line with the reader expectations.*

L348-353; "More generally, although our focus has been on evaluating salinity levels with regards to irrigation-specific thresholds, our results and salinity dataset could be used for other impact assessments where salinity is known to be a risk factor. This includes ecological assessments, since high freshwater salinity levels are known to cause multiple effects on biota, with impacts on species richness, abundance and functional traits, as well as on ecosystem processes^{6,45}."

5) Conductivity is a continuous variable. I don't fully understand why Wilcoxon tests between salinity classes are used to statistically assess the different contribution of each driver. Why not building linear mixed models (with the sub-basin identity as a random factor) or GAMs to assess the relationship between conductivity and each driver? This is partially addressed with Random Forests in the manuscript, but I still think that building models without categorising the sites could help to understand the relationship between the drivers and the conductivity values observed.

→ *The Wilcoxon rank sum test (Synonymous: Mann-Whitney test, Mann-Whitney U test, Wilcoxon-Mann-Whitney test and two-sample Wilcoxon test) is a non-parametric alternative to the t-test for comparing two means. It's particularly recommended in a situation where the data are not normally distributed. We used the Wilcoxon test for comparing the mean values of each considered driver between 2 salinity impact groups, to see if there is any significant shift in the driver level between any two salinity classes (Low-High, Low-Moderate, Moderate-High). Using this test, we can statistically analyse if the contribution of a driver differs markedly between regions of low and high salinity levels, which is then an indication of their respective contributions to observed salinity levels. This type of test has been used extensively in similar applications as our study, for example to assess the contribution of different human-drivers on changes in water balance components across 100 hydrological basins worldwide (Jaramillo and Destouni 2015). In that study, the results of the Wilcoxon rank sum test (showing a statistically higher level of evapotranspiration in highly modified basins compared to low modified basins) were used to infer the effect of human impacts on global hydrological changes. Similarly, in our study, we use these results as an indication of the contribution of the different drivers to observed salinity changes across different salinity impact classes (Low, Moderate, High).*

The aim of the Random Forest approach was then to further investigate the influence of the considered drivers on observed salinity levels. We do use the model to predict individual salinity values (long term EC for each site) and the categorisation is only regarding the 2 models; one including all sub-basins that have shown increasing salinisation and one for all sub-basins that have shown decreasing salinity trends. We used this division due to the hypothesis that there could be different drivers influencing basins undergoing salinisation vs freshening – and we wanted to investigate this further. This type of approach also allows us to bring additional value that could complement, but still expand on the analysis described above (Fig. 4).

We do believe there are several approaches that could have been explored for our study. However, the main reason to why we chose RF over e.g. LMMs or GAMs, is because RF is a machine-learning technique which can model nonlinear relationships (e.g. Gacto et al. 2019), are insensitive to over-fitting and works well even if predictor variables are dependent (multi-collinearity), which is problematic for LMMs and GAMs (e.g. Harrison et al., 2018, Zhang and Chen, 2013). RF have also been shown to have high predictive performance compared with other machine-learning approaches and are increasingly being used in environmental applications, including salinity-specific studies (e.g. Olson 2019, Estévez et al., 2019, Southworth et al., 2018). In contrast to linear

parametric models (like LMMs), data-driven machine learning methods (like RF) can be applied to derive insight from the raw data without a priori assumptions, due to the non-parametric nature of decision-trees (Ngufor et al. 2019). Due to this, RF models can capture non-linear effects and complex interactions, which often leads to a higher prediction accuracy than traditional regression algorithms. An additional advantage of RF models is that they can handle a large number of potential predictor variables, which was useful in our application. Another important reason to why we chose RF, is that it could be used together with the newly improved CPI method (Debeer and Strobl 2020). The applied CPI method has been shown to work very well in RF applications which are more focused on understanding the relative contribution of predictor variables (i.e. our considered drivers), rather than just aiming for as good as possible "black-box" type of prediction. CPI adds interpretability to the model by assessing the relevance of predictor variables in predicting an outcome and are becoming more used in applications aiming to move towards more interpretable machine learning outcomes, which is in line with our study aim and approach.

6) I wonder if the salinity trends could be broken into seasonal trends, with the salinisation trend (i.e. decreasing or increasing) depending on the season due to climatic factors and irrigation schemes.

→ We agree that seasonality could have a large impact, particularly on the salinity levels (as shown in Fig. 3b) but also on the trends. However, the objective of this first assessment was to focus more on the large-scale, long-term evaluations across many sub-basins. As a first cross-regional analysis of salinity levels, trends and drivers, we chose the 30-year period, which enables long-term assessment of both salinity changes and its relation to various drivers (including climate parameters). However, the data gaps within our selection criteria (up to 15 years within the considered 30-year period of 1980-2010) do not allow for season-specific trend analysis, which is why the focus on long-term annual averages are most suitable for this study. However, to highlight the value of accounting for seasonal aspects, we have now mentioned this in the discussion as an important aspect to consider in further research efforts;

L327-335; "However, our results also reveal large spatial heterogeneity in the direction of trends, both within and across (regional) river (sub-)basins. We also found that many sub-basins which did not exhibit persistent salinity problems, still temporarily exceeded salinity irrigation water-use threshold levels. This suggests that problems of freshwater salinisation might fluctuate highly, both within and across seasons⁴², which could have important implication for intra-annual irrigation water use. In addition, salinity water use constraints are underestimated if quantified at the annual to long-term scales, highlighting the need for further studies to account for seasonal aspects with regards to both salinity levels and trends."

Abstract

7) There are some papers analysing the causes of salinisation at regional scales (10–14). Thus, I am not sure if stating that "its drivers across regions are largely unknown", or it is more accurate to say that the knowledge is restricted to some regions in Australia, Europe and the US.

→ As mentioned in our response to comment 1, we have now updated the abstract (and also title) to better reflect the focus of salinity impacts on irrigation water use constrains and agricultural activities as being the focus of human drivers. In light of this, the first sentence now reads: "Freshwater salinisation is a growing problem, yet cross-regional assessments of freshwater salinity status and the impact of agricultural and other sectoral uses are lacking". This narrows down the focus, and is still a valid sentence, since assessments of agricultural drivers on salinity levels and trends have not been the focus of any cross-regional assessments (as is now also made clearer in the introduction, including the objectives;

L57-62; "With regards to agricultural drivers, there are several studies at local and regional scales^{22,23}, including raising saline groundwater due to irrigation withdrawals²⁴, or spreading of salt-containing fertilisers to the groundwater and/or surface water system²⁵. However, there is a lack of systematic assessments of the impact of agricultural activities on surface water salinisation at cross-regional to global scales."

8) What do you mean by "consistent effects of higher contribution"? This is rather vague; you could be more specific.

→ This has now been re-formulated to (L147-155): "Our results show higher contributions of (human) agricultural drivers in sub-basins of Moderate to High salinity levels (Fig. 4; yellow and orange boxplots), compared to those of Low salinity levels (Fig. 4; green boxplots). These results are particularly strong for irrigation-related variables, with significantly higher levels of irrigation water withdrawals (Irr. ww) and irrigation return flows (Irr. rf) in sub-basins within the High, compared to the Low salinity impact class (Fig. 4; green-orange boxplots, and Wilcoxon test results in Supplementary Table 1; Low-High pair comparison)."

We hope that this is now clearer.

Introduction

9) L26-27 This is a very general statement. I don't think it is needed.

→ We have removed this sentence.

10) L49 The models built by Le et al. (12) for Germany and Olson (13) for the US, indirectly provide information on the drivers of salinisation (i.e. more robust predictor variables have a stronger contribution to salinisation) that could be taken into account.

→ The modelling study of Le et al. considers hydro-climatic drivers and we have included a reference to this study in the introduction;

L53-55: "In addition to hydroclimatic drivers¹⁷ (**Le et al., 2019**), relevant human drivers to inland freshwater salinisation include road salt application¹⁸, mining¹⁹ and agriculture²⁰."

The study by Olson 2018 is now explicitly mentioned in the discussion;

L362-367: "These results both agree with, and increase the spatial extent of previously reported effects of agricultural activities on salinisation^{21,22}. For example, a study using a similar RF approach as here and considering drivers to river salinisation across the US showed that the largest salinity increases were impacted by a combination of agriculture (% crop land cover), mining and groundwater pumping²⁹ (Olson 2018)."

11) L154 In fact more vegetation could mean less salt in the water because the salt uptake by the roots.

→ We agree that root uptake can be a plausible process in which salt could be removed. However, as pointed out by Reviewer #1 (comment 2), our contra-intuitive finding of higher salinity in lower aridity regions needed further consideration and our originally proposed explanation of vegetation effects on salinisation in lower aridity regions may not be robust enough. The hypothesis and analyses with regards to vegetation has now been removed, and replaced by analyses and explanations with regards to aridity and irrigation water use and activities, as described in detail in our response to reviewer #1 above and exemplified for several instances in the manuscript where these changes have been implemented and discussed.

12) L161 This makes sense. In naturally saline streams the vegetation is usually dominated by shrubs (7).

→ As mentioned above, we have now removed the vegetation part and believe we can better explain the elevated salinity levels in lower aridity regions due to higher irrigation activities in less arid sub-basins, rather than with vegetation patterns (see our detailed responses to reviewer #1, comment 2).

13) Figure 1: I miss road salt application and resource extraction. Also, naturally saline groundwaters can be a significant driver of salinization in some regions (15). Finally, I am not sure how dams act as a driver of salinization; this might need to be explained.

→ Please see our previous response to your comment nr 1, where we now have done additional analyses with regards to mining and road salt applications, as well as clarified the focus of agricultural activities. For the general overview schematic in Fig. 1, we have now however also included road salt and resource extractions, as well as a clearer representation of groundwater flows into the updated figure. We have clarified in the Supplementary Materials (S2) why we consider dams and reservoirs as agriculturally relevant drivers, reading; "Reservoirs are an important water supply source for irrigation¹⁴ (Yoshikawa et al., 2014) and could contribute to increasing freshwater salinity levels through evapo-concentrations effects caused by increased relative evapotranspiration from the open water surfaces of reservoirs¹⁵ (Jaramillo and Destouni 2015)."

14) Figure 4: are all the boxplots showing statistically significant differences between classes? For some of them I see very weak differences.

→ *Not all boxplots show statistically significant differences, but in the previous version, the reader was referred to the SM to look up which drivers did show significant differences between classes. However, we have now updated figure 4 (see updated figure in the manuscript file and above in our detailed response to reviewer #1, comment 2), with a * to show where there is a statistically significant difference in driver contributions between salinity classes (between low-high groups, in accordance with the results of SM Table 1). We hope that these modifications will make the results of Fig. 4 easier for the reader to interpret and without having to move to the SM to look this up.*

Miguel Cañedo-Argüelles

References

1. W. D. Hintz, R. A. Relyea, A review of the species, community, and ecosystem impacts of road salt salinisation in fresh waters. *Freshw. Biol.* 64, 1081–1097 (2019).
2. S. S. Kaushal et al., Increased salinization of fresh water in the northeastern United States. *Proc. Natl. Acad. Sci.* 38, 13517–13520 (2005).
3. J. Bäche, E. Coring, Biological effects of anthropogenic salt-load on the aquatic Fauna: A synthesis of 17 years of biological survey on the rivers Werra and Weser. *Limnol. - Ecol. Manag. Inl. Waters.* 41, 125–133 (2011).
4. M. A. Palmer et al., Mountaintop mining consequences. *Science* (80-.). 327, 148–149 (2010).
5. R. D. Vidic, S. L. Brantley, J. M. Vandebossche, D. Yoxtheimer, J. D. Abad, Impact of shale gas development on regional water quality. *Science.* 340, 1235009 (2013).
6. C. Gutiérrez-Cánovas et al., Do all roads lead to Rome? Exploring community trajectories in response to anthropogenic salinization and dilution of rivers. *Philos. Trans. R. Soc. B Biol. Sci.* 374 (2019) (available at <http://rstb.royalsocietypublishing.org/content/374/1764/20180009.abstract>).
7. A. Millán et al., Mediterranean saline streams in southeast Spain: What do we know? *J. Arid Environ.* 75, 1352–1359 (2011).
8. M. Cañedo Argüelles et al., Salinisation of rivers: an urgent ecological issue. *Environ. Pollut.* 173, 157–167 (2013).
9. Hart, et al, A review of salt sensitivity of Australian freshwater biota. *Hydrobiologia.* 210, 105–144 (1991).
10. S. S. Kaushal et al., Freshwater salinization syndrome on a continental scale. *Proc. Natl. Acad. Sci.* 115, E574–E583 (2018).
11. E. Estévez, T. Rodríguez-Castillo, M. G.-F. Alexia, M. Cañedo-Argüelles, J. Barquín, Drivers of spatio-temporal patterns of salinity in Spanish rivers: a nationwide assessment. *Philos. Trans. R. Soc. B Biol. Sci.* 374, 20180022 (2019).
12. T. D. H. Le et al., Predicting current and future background ion concentrations in German surface water under climate change. *Philos. Trans. R. Soc. B Biol. Sci.* 374, 20180004 (2019).
13. J. R. Olson, Predicting combined effects of land use and climate change on river and stream salinity. *Philos. Trans. R. Soc. B Biol. Sci.* 374, 20180005 (2019).
14. G. B. Allison et al., Land clearance and river salinisation in the western Murray Basin, Australia.

J. Hydrol. 119, 1–20 (1990).

15. C. Li, X. Gao, S. Li, J. Bundschuh, A review of the distribution, sources, genesis, and environmental concerns of salinity in groundwater. *Environ. Sci. Pollut. Res.*, 1–18 (2020).

Additional references included in the author response to reviewer #2:

Buvaneshwari, S. et al. Potash fertilizer promotes incipient salinization in groundwater irrigated semi-arid agriculture. *Scientific Reports* 10, 3691 (2020).

Barros, R., Isidoro, D. & Aragüés, R. Three study decades on irrigation performance and salt concentrations and loads in the irrigation return flows of La Violada irrigation district (Spain). *Agriculture, Ecosystems & Environment* 151, 44–52 (2012).

Kass, A., Gavrieli, I., Yechieli, Y., Vengosh, A. & Starinsky, A. The impact of freshwater and wastewater irrigation on the chemistry of shallow groundwater: a case study from the Israeli Coastal Aquifer. *Journal of Hydrology* 300, 314–331 (2005).

Maus, V. et al. A global-scale data set of mining areas. *Scientific Data* 7, 289 (2020).

Bock, A.R., Falcone, J.A., and Oelsner, G.P., 2018, Estimates of Road Salt Application across the Conterminous United States, 1992–2015: U.S. Geological Survey data release.

Kaushal, S. S. et al. Freshwater salinization syndrome on a continental scale. *PNAS* 115, E574–E583 (2018).

Kelly, W. R., Panno, S. V. & Hackley, K. C. Impacts of Road Salt Runoff on Water Quality of the Chicago, Illinois, Region, Road Salt Impacts, Chicago. *Environmental and Engineering Geoscience* 18, 65–81 (2012).

Jaramillo, F. & Destouni, G. Local flow regulation and irrigation raise global human water consumption and footprint. *Science* 350, 1248–1251 (2015).

Gacto, M. J., Soto-Hidalgo, J. M., Alcalá-Fdez, J. & Alcalá, R. Experimental Study on 164 Algorithms Available in Software Tools for Solving Standard Non-Linear Regression Problems. *IEEE Access* 7, 108916–108939 (2019).

Harrison, X. A. et al. A brief introduction to mixed effects modelling and multi-model inference in ecology. *PeerJ* 6, (2018).

Zhang, G. & Chen, J. J. Adaptive Fitting of Linear Mixed-Effects Models with Correlated Random-effects. *J Stat Comput Simul* 83, (2013).

Olson, J. R. Predicting combined effects of land use and climate change on river and stream salinity. *Philosophical Transactions of the Royal Society B: Biological Sciences* 374, (2019).

Estévez, E., Rodríguez-Castillo, T., González-Ferreras, A. M., Cañedo-Argüelles, M. & Barquín, J. Drivers of spatio-temporal patterns of salinity in Spanish rivers: a nationwide assessment. *Philosophical Transactions of the Royal Society B: Biological Sciences* 374, 20180022 (2019).

Southworth, J. et al. Using a coupled dynamic factor – random forest analysis (DFRFA) to reveal drivers of spatiotemporal heterogeneity in the semi-arid regions of southern Africa. *PLOS ONE* 13, e0208400 (2018).

Ngufor, C., Van Houten, H., Caffo, B. S., Shah, N. D. & McCoy, R. G. Mixed effect machine learning: A framework for predicting longitudinal change in hemoglobin A1c. *Journal of Biomedical Informatics* 89, 56–67 (2019).

Debeer, D. & Strobl, C. Conditional permutation importance revisited. *BMC Bioinformatics* 21, 307 (2020).

Tangdamrongsub, N. et al. Improving estimates of water resources in a semi-arid region by assimilating GRACE data into the PCR-GLOBWB hydrological model. *Hydrology and Earth System Sciences* 21, 2053–2074 (2017).

Thorslund, J. & van Vliet, M. T. H. A global dataset of surface water and groundwater salinity measurements from 1980–2019. *Scientific Data* 7, 231 (2020).

REVIEWERS' COMMENTS

Reviewer #1 (Remarks to the Author):

Comments on Revision of Thorslund et al. Nature Comms, May 2021.
Common irrigation drivers of freshwater salinisation in river basins worldwide

In my original review of this manuscript, I found the submission to be strong but with specific issues that needed to be clarified for accuracy. In particular, I had concerns with an unexpected finding related to salinity and the aridity index (PET/P). The authors have addressed my comments pertaining to the issue of aridity and salinization by clarifying (and in some cases removing) suspect claims, and by investigating the issue further and presenting a more detailed perspective on the relationship between salinization and aridity. More broadly, I find the author response and updated manuscript to have thoroughly addressed each of the comments I presented in the initial review.

After re-reading the manuscript, the submission remains strong. The authors compile a broad group of datasets to assess freshwater salinization through an inter-regional comparison and demonstrate the importance of irrigation as a driver of salinization, as well as identify a range of drivers that contribute to reductions in salinity. This represents an important advancement that uses empirical analysis to integrate regional information into a more general understanding of freshwater salinization processes. The manuscript is well-written with methods and findings clearly articulated. I recommend the manuscript for publication.

Reviewer #2 (Remarks to the Author):

The authors have done an excellent job in addressing my comments. I want to congratulate them. I am really looking forward to seeing this manuscript published. I think that it will attract a lot of attention since it provides the most comprehensive analysis of the relative importance of the different drivers of freshwater salinization ever performed. I only have some minor comments:

My comment number 5 in my previous review was related to the fact that conductivity is a continuous variable and there is no need for categorizing it (as the authors do in this paper, using the low, moderate, and high categories). I understand and approve the statistical methods used, I was just wondering if categories were needed. Anyhow, I understand that the authors will stick to this approach and it is ok with me.

I am still a bit concerned about the lack of clear differentiation between natural salinity and anthropogenic salinization. I would emphasize that the "overall salinity impact classes" refer to salinity levels that make water unsuitable for irrigation of most crops (although salt-tolerant crops also exist! especially in arid regions), but it does not necessarily mean that this salinity is caused by human impacts. Also, saline ecosystems are valuable and widespread in arid regions. I would at least mention this in the discussion, so the reader does not get the impression that high salinity equals water pollution and ecosystem degradation in all cases.

Figure 1: in my opinion, the road salt application should be placed in the mountains, close to the snow.

Manuscript Number: NCOMMS-20-50715A

Manuscript Title: Common irrigation drivers of freshwater salinisation in river basins worldwide

Corresponding Author: Dr Josefin Thorslund

We would again like to thank both reviewers for going through the manuscript revisions and for their final comments. Please find our point-by-point responses to the remaining comments below.

Yours sincerely,

Josefin Thorslund

Responses to Reviewer #1 comments

Reviewer #1 (Remarks to the Author):

Comments on Revision of Thorslund et al. Nature Comms, May 2021.

Common irrigation drivers of freshwater salinisation in river basins worldwide

1) In my original review of this manuscript, I found the submission to be strong but with specific issues that needed to be clarified for accuracy. In particular, I had concerns with an unexpected finding related to salinity and the aridity index (PET/P). The authors have addressed my comments pertaining to the issue of aridity and salinization by clarifying (and in some cases removing) suspect claims, and by investigating the issue further and presenting a more detailed perspective on the relationship between salinization and aridity. More broadly, I find the author response and updated manuscript to have thoroughly addressed each of the comments I presented in the initial review.

After re-reading the manuscript, the submission remains strong. The authors compile a broad group of datasets to assess freshwater salinization through an inter-regional comparison and demonstrate the importance of irrigation as a driver of salinization, as well as identify a range of drivers that contribute to reductions in salinity. This represents an important advancement that uses empirical analysis to integrate regional information into a more general understanding of freshwater salinization processes. The manuscript is well-written with methods and findings clearly articulated. I recommend the manuscript for publication.

→ *We are happy to hear that we have adequately addressed each of your initial review comments. Thank you again for your time and providing constructive review comments that helped improve our manuscript substantially.*

Responses to Reviewer #2 comments

Reviewer #2 (Remarks to the Author):

The authors have done an excellent job in addressing my comments. I want to congratulate them. I am really looking forward to seeing this manuscript published. I think that it will attract a lot of attention since it provides the most comprehensive analysis of the relative importance of the different drivers of freshwater salinization ever performed. I only have some minor comments:

1) My comment number 5 in my previous review was related to the fact that conductivity is a continuous variable and there is no need for categorizing it (as the authors do in this paper, using the low, moderate, and high categories). I understand and approve the statistical methods used, I was just wondering if categories were needed. Anyhow, I understand that the authors will stick to this approach and it is ok with me.

→ Thank you for these encouraging comments on our revised manuscript. We are happy to hear that the statistical methods used are now understandable and considered appropriate. As mentioned previously, the categorization was one way to assess the impact of salinity levels in terms of limitations for irrigation water use, which was judged appropriate for this approach based on common global salinity ranges of irrigation water use thresholds, whereas for the predictive aspects of the Random Forest model, we used continuous conductivity values.

2) I am still a bit concerned about the lack of clear differentiation between natural salinity and anthropogenic salinization. I would emphasize that the "overall salinity impact classes" refer to salinity levels that make water unsuitable for irrigation of most crops (although salt-tolerant crops also exist! especially in arid regions), but it does not necessarily mean that this salinity is caused by human impacts. Also, saline ecosystems are valuable and widespread in arid regions. I would at least mention this in the discussion, so the reader does not get the impression that high salinity equals water pollution and ecosystem degradation in all cases.

→ We have now updated the discussion to better reflect that our salinity classification and assessments relates to irrigation limitations for most crops. We also brought in aspects of salt-tolerant crops, as well as the value of natural saline ecosystems (see updated text sections below, as well as the 2 new references). We think that the discussion now better acknowledges that high salinity can have a range of impacts (both positive and negative) and causes, which needs to be acknowledged, depending on the focus of the assessment.

L266-272: "Yield reductions of between 10-25 % have been shown to affect saline sensitive crops (e.g. beans, carrots and lettuce) when irrigating with saline water of around 1000 $\mu\text{S cm}^{-1}$ (i.e. salinity levels within the Moderate and High salinity impact classes of this study)⁴³.

Although salt-tolerant crops (such as halophytes) exist and may provide a suitable adaptation strategy to high salinization for some regions, the large-scale potential of saline irrigation is still an underexplored research area⁴⁴. In addition, risks of higher yield reductions increase over time, due to soil salinity build-up⁴⁵."

L277-289: "More generally, although our focus has been on evaluating salinity levels with regards to limitations for irrigation of most crops, our results and salinity dataset could be used for other impact assessments. This includes other human water uses, such as increasing water scarcity for the domestic and industrial sectors⁴⁶, as well as human health⁴⁷. Also, in terms of ecological assessments, high freshwater salinity levels can have multiple effects on biota, with impacts on species richness, abundance and functional traits, as well as on ecosystem processes^{6,48}. However, it is worth noting that naturally occurring saline aquatic systems, especially in arid regions, also provide important ecosystem services and valuable habitats for salt adapted species⁴⁹. Therefore, high salinity levels do not always result in ecosystem degradation, but salinity levels must be evaluated with regards to underlying conditions and specific impact assessments. Further cross-regional assessments with regards to these various aspects could add valuable knowledge to the more wide-ranging impacts of the here quantified salinity levels."

New references:

44. Rozema, J. & Flowers, T. Crops for a Salinized World. *Science* 322, 1478–1480 (2008).

49. Zadereev, E. et al. Overview of past, current, and future ecosystem and biodiversity trends of inland saline lakes of Europe and Central Asia. *Inland Waters* 10, 438–452 (2020).

3) Figure 1: in my opinion, the road salt application should be placed in the mountains, close to the snow.

→ Figure 1 has now been updated with regards to moving the road salt application schematic closer to the snowy region.